# LouisKV: Efficient KV Cache Retrieval for Long Input-Output Sequences

**Wenbo Wu[1],[*] Qingyi Si[2],[*] Xiurui Pan[1], Ye Wang[3], Jie Zhang[1],[†]**

[1]Peking University    [2]Huawei Technologies Ltd.
[3]Chongqing University of Post and Telecommunications

## Abstract

While Key-Value (KV) cache succeeds in reducing redundant computations in auto-regressive models, it introduces significant memory overhead, limiting its practical deployment in long-sequence scenarios. Existing KV retrieval methods attempt to mitigate this by dynamically retaining only a subset of KV entries on the GPU. However, they still suffer from notable efficiency and accuracy bottlenecks due to per-token retrieval and coarse-grained page-level KV management strategy, especially in long-output reasoning scenarios. With the emergence of large reasoning models, efficiently handling such scenarios has become increasingly important. To address this issue, we present two key observations: (1) critical KVs exhibit strong temporal locality during decoding, and (2) these KVs exhibit distinct distribution patterns across the input prompt and the generated output. Building on these observations, we propose *LouisKV*, an efficient KV cache retrieval framework designed for various long-sequence scenarios. Specifically, LouisKV introduces a semantic-aware retrieval strategy that leverages temporal locality to trigger retrieval only at semantic boundaries, drastically reducing computation and data transfer overhead. LouisKV also designs a decoupled, fine-grained management scheme that tailors differentiated strategies for input and output sequences to create retrieval units that better match the model's attention patterns, thereby enabling the precise identification of critical KVs. Furthermore, to boost system efficiency, LouisKV incorporates several kernel-level optimizations, including custom Triton and CUDA kernels to accelerate the KV clustering and retrieval. Evaluation results show that LouisKV achieves up to $4.7\times$ speedup over state-of-the-art KV retrieval methods while maintaining near-lossless accuracy across diverse long-sequence tasks, including long-input short-output, short-input long-output, and long-input long-output scenarios.

## 1 Introduction

Large language models (LLMs) (Achiam et al., 2023; Guo et al., 2025) have demonstrated remarkable capabilities in both long-context understanding for tasks like long document question answering (Bai et al., 2023), and in long chain-of-thought (CoT) generation for complex reasoning tasks like mathematics (Hendrycks et al., 2021). The inference process of mainstream LLMs is auto-regressive, with the KV cache (Shi et al., 2024) stored in memory to avoid recomputation. However, the memory footprint of the KV cache grows approximately linearly with the sequence length, easily exceeding GPU memory capacity.

To address this challenge, recent studies have proposed sparse attention mechanisms to reduce KV cache usage. One line of work, known as KV dropping (Xiao et al., 2023; Zhang et al., 2023; Feng et al., 2024), retains only important tokens and permanently discards those deemed less relevant. However, such methods fail to account for the dynamic nature of KV importance, that is, tokens initially considered unimportant may become critical in subsequent steps, leading to reduced inference accuracy. To address this, another line of work, KV retrieval (Chen et al., 2024a; Tang et al., 2024;

---

[*]Equal contribution.
[†]Corresponding author.

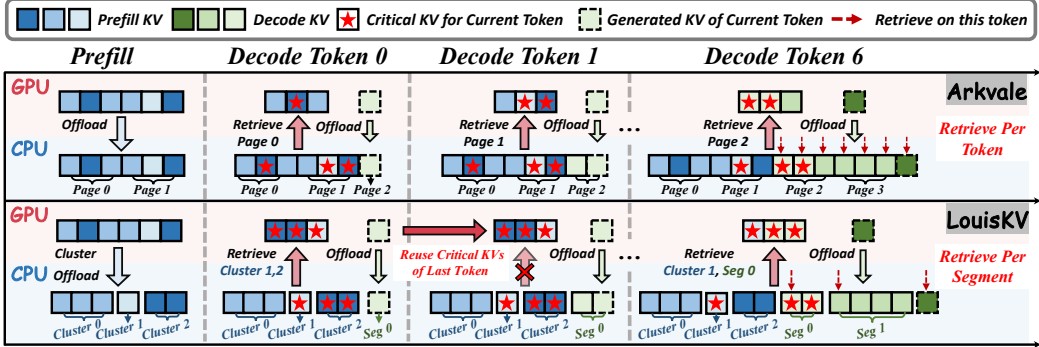

Figure 1: Comparison of Arkvale and LouisKV. Arkvale adopts page-level KV management and retrieves critical pages from the CPU for every decoding token, leading to high transfer overhead and potential accuracy degradation. In contrast, LouisKV reuses critical KVs by exploiting temporal locality to significantly reduce retrieval frequency. It also employs a decoupled KV management scheme, enabling precise retrieval to improve transfer efficiency while maintaining high accuracy.

Liu et al., 2024a; Chen et al., 2024b; Pan et al., 2024), has emerged as a promising alternative. This approach preserves the full KV cache, typically in CPU memory to avoid GPU Out-Of-Memory (OOM) errors, whereas dynamically estimating a subset of important entries and retrieving them for the current token's generation.

Although existing KV retrieval methods achieve better accuracy than KV dropping approaches by retaining complete information, these methods still face significant bottlenecks in both inference efficiency and retrieval performance. As shown in Figure 1, a primary issue is that existing methods (Zhang et al., 2025; Lee et al., 2024; Xu et al., 2025) trigger retrieval at every decoding token. This introduces two primary sources of overhead: the computational cost of selecting critical tokens and the data transfer latency of recalling KV entries from the CPU. This accumulated overhead becomes particularly prohibitive in scenarios involving long-output reasoning models (Guo et al., 2025), which have become a mainstream paradigm. Meanwhile, existing methods (Chen et al., 2024a; Xiao et al., 2024; Yuan et al., 2025) typically adopt page-level KV management strategies. However, this coarse-grained management strategy frequently transfers entire pages containing numerous non-critical KV entries, which either increase data transfer overheads or reduce inference accuracy. Therefore, enhancing inference efficiency while preserving accuracy across all long-sequence scenarios has become a key challenge.

To address this bottleneck, we conduct an empirical analysis of critical KV access patterns in long-output reasoning models (Yang et al., 2025). Our two key observations bring new insights to well address two fundamental questions: *how to trigger retrieval to reduce overhead*, and *how to manage the KV cache to improve retrieval precision*. First, we observe that *critical KVs access exhibits strong temporal locality*. Specifically, the sets of critical KV entries accessed in adjacent decoding tokens show high similarity. For instance, during mathematical reasoning, the model produces intermediate reasoning steps wherein tokens within the same step consistently attend to the same mathematical lemmas. This insight motivates triggering retrieval only at semantic boundaries. Second, we observe that *critical KVs exhibit distinct distribution patterns across input and output sequences*. Specifically, for the currently generated token, critical KVs in long input sequences are often distributed sparsely throughout the context, while those in long output sequences of reasoning models tend to concentrate locally within some reasoning steps. This inspires the adoption of different KV management strategies for the prefill and decode stages.

Building on these observations, we propose *LouisKV*, an efficient KV cache retrieval framework that integrates a semantic-aware retrieval strategy with a decoupled fine-grained management scheme, as shown in Figure 1. To exploit *temporal locality* during decoding, LouisKV groups consecutive decoding tokens into coherent segments based on query vector similarity. It triggers retrieval only at segment boundaries to drastically reduce the retrieval overhead. Based on the attention distribution, LouisKV proposes a decoupled KV management scheme tailored for distinct distribution patterns in the input and output sequence. Specifically, for the long input sequence, LouisKV employs clustering to group KV entries of semantically similar tokens into *semantic clusters*; for the long output

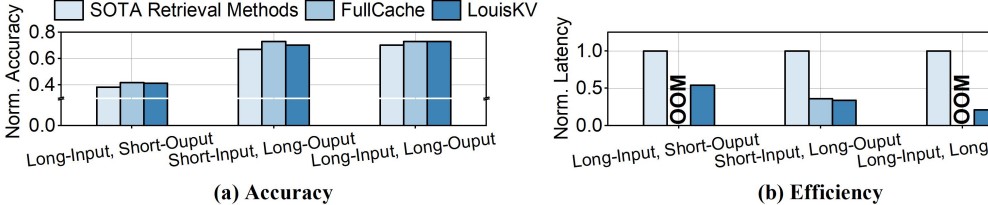

Figure 2: Accuracy and efficiency comparison on various long-sequence tasks. (a) LouisKV achieves accuracy comparable to FullCache (the lossless baseline) and superior to state-of-the-art retrieval methods. (b) LouisKV significantly reduces inference latency, substantially outperforming state-of-the-art retrieval methods, while also avoiding the Out-Of-Memory errors that FullCache faces in long-sequence and large batch size scenarios.

sequence, it leverages dynamic boundaries to partition the KV cache into *temporal segments*. By creating retrieval units that better match attention distributions, LouisKV enables precise identification and transfer of critical KV entries, achieving higher accuracy and lower transfer overheads than page-level methods. Furthermore, to boost system efficiency, LouisKV incorporates several kernel-level optimizations, including custom Triton and CUDA kernels to accelerate KV clustering and retrieval. As validated by extensive experiments summarized in Figure 2, LouisKV maintains near-lossless accuracy on various long-sequence tasks while achieving up to a 4.7× speedup in end-to-end latency compared to state-of-the-art retrieval methods.

Our contributions are as follows: (1) We conduct an empirical analysis of critical KV access patterns, revealing two key observations: strong temporal locality and distinct distribution patterns. (2) Based on these insights, we propose LouisKV, a novel KV retrieval framework that incorporates a semantic-aware retrieval strategy and a decoupled management scheme. To our knowledge, LouisKV is the first KV retrieval framework designed to comprehensively cover diverse long-sequence tasks. (3) We extensively evaluate LouisKV on various LLM benchmarks, demonstrating that it achieves a 4.7× speedup in end-to-end latency while maintaining near-lossless accuracy, as shown in Figure 2.

## 2 RELATED WORK

**Scaling LLMs to Long Contexts and Generations.** The capability of LLMs to process long-sequence tasks is rapidly advancing. On the one hand, modern models have supported vastly longer inputs. For instance, Qwen3 (Yang et al., 2025) can handle contexts of 128K tokens, while Gemini (Team et al., 2023) models push the boundary to 2M tokens. On the other hand, a parallel trend involves generating longer outputs. Driven by test-time scaling laws, prompting models to generate more extensive, step-by-step reasoning at inference has proven to be a cost-effective method to boost model performance (Guo et al., 2025; OpenAI, 2024; Yang et al., 2025). Our work, LouisKV, targets KV cache optimization in both long-input and long-output scenarios.

**KV Cache Dropping.** To mitigate GPU memory pressure, several methods (Zhang et al., 2023; Hu et al., 2025; Chen et al., 2025; Cai et al., 2026) employ strategies that permanently discard KV entries based on attention scores or heuristics rules. However, these approaches lead to irreversible information loss, as discarded KV entries cannot be recalled for subsequent decoding steps. Even more advanced techniques like Scope (Wu et al., 2025), which attempt to preserve context by retaining compressed prefill tokens, still fundamentally rely on eviction mechanisms that risk discarding critical data. Consequently, while these methods are computationally efficient, they suffer from significant accuracy degradation in long-output reasoning tasks.

**KV Cache Retrieval and Optimization.** To avoid the irreversible information loss inherent in KV dropping methods, KV retrieval approaches preserve the full KV cache in CPU memory and dynamically fetch a critical subset for computation. However, existing methods still suffer from severe efficiency and accuracy bottlenecks. First, as illustrated in Figure 1, triggering retrieval operations for every single decoding token introduces prohibitive computational and data transfer overheads, particularly in long-output scenarios. Second, traditional coarse-grained management (e.g., page-level) results in the transfer of numerous non-critical tokens, which exacerbates bandwidth waste and can even degrade model accuracy (a detailed bottleneck analysis is provided in Appendix A).

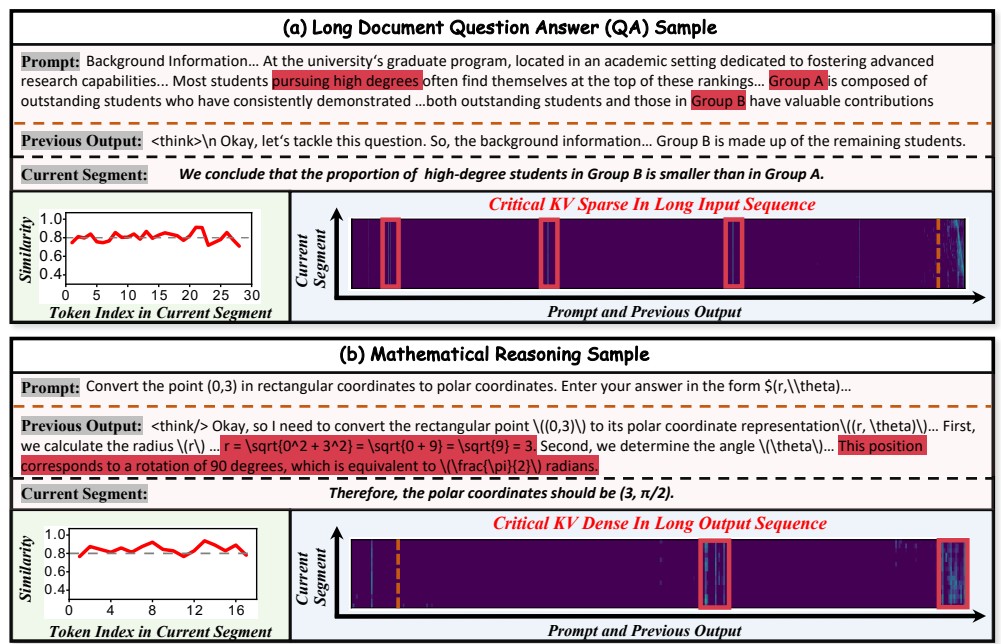

Figure 3: Access patterns of critical KVs in different long-sequence tasks. First, both tasks demonstrate temporal locality: during the generation of a coherent segment (Current Segment), the similarity of critical KV sets maintains high values. Second, they reveal a distinct spatial distribution: (a) in a long-document task, attention is sparsely distributed in the prompt, whereas (b) in a mathematical reasoning task, attention is densely focused on some intermediate steps in the previous output.

While recent works such as ClusterKV (Liu et al., 2024b), Squeezed Attention (Hooper et al., 2025), and SentenceKV (Zhu et al., 2025) attempt to improve retrieval granularity via semantic clustering or syntactic boundaries, they predominantly focus on static input sequences, neglecting the dynamic attention patterns inherent in long-output generation. In contrast, LouisKV addresses these fundamental limitations through two targeted innovations: an adaptive semantic-aware retrieval strategy that mitigates per-token overhead by exploiting temporal locality, and a decoupled fine-grained management scheme tailored for both input sparsity and output density.

## 3 MOTIVATION

### 3.1 PRELIMINARIES

The auto-regressive inference of LLMs involves two phases: prefill and decode. During prefilling, the model processes the entire input prompt to generate initial KV cache for each layer, denoted $K_{inp} \in \mathbb{R}^{P \times d}$ and $V_{inp} \in \mathbb{R}^{P \times d}$, where $P$ is the prompt length and $d$ is the hidden dimension. During decoding, the model generates tokens auto-regressively until encountering an end-of-sequence token. At decoding step $t$, the model generates a query vector $q_t \in \mathbb{R}^{1 \times d}$ for each layer. Simultaneously, the corresponding key $k_t$ and value $v_t \in \mathbb{R}^{1 \times d}$ are computed and appended to the historical KV cache, forming updated $K_t$ and $V_t$. The output $o_t$ is then computed via the attention mechanism:

$$o_t = A_t V_t = \text{Softmax}\left(\frac{q_t K_t^T}{\sqrt{d}}\right) V_t$$

Here, $A_t$ represents the attention weights indicating the importance of each historical KV entry to the current query $q_t$.

KV retrieval methods usually select a subset of KV entries under a fixed budget $B$. Quest (Tang et al., 2024) and Arkvale (Chen et al., 2024a) partition the entire key cache $(k_0, k_1, \ldots, k_{P+t})$ into $m$ fixed-size pages and construct index $C_i$ for each page $i$. At every decoding step $t$, they select the most critical pages by computing the similarity between the current query $q_t$ and all indices $(C_0 \ldots C_{m-1})$. All KV entries within a selected page are retrieved for the attention computation. The

selection process can be formalized as $\mathcal{I}_t = \text{Sel}(q_t, K_t)$, where $\mathcal{I}_t$ is the set of KV indices selected in step $t$, with the constraint $|\mathcal{I}_t| \leq B$. Ultimately, only the selected KV subset $(K_{\mathcal{I}_t}, V_{\mathcal{I}_t})$ participates in the attention computation, significantly reducing the inference latency and memory footprint.

## 3.2 OBSERVATIONS

We conducted an in-depth analysis of critical KV access patterns on long-sequence tasks (Jia, 2025; Ling et al., 2025), leading to two key observations that directly motivate our design. These observations provide insights into two fundamental questions: 1) When should retrieval be triggered, and 2) How should the KV cache be managed to enable precise and efficient retrieval?

**Observation 1: Critical KVs access exhibits strong temporal locality during decoding.** The model tends to produce its output as a sequence of semantically coherent *segments*. For instance, one reasoning step in a mathematical problem forms one segment, as illustrated in Figure 3(b). We hypothesize that within each segment, the model continuously attends to a relatively similar subset of tokens to maintain semantic coherence.

To validate this hypothesis, we calculated the Jaccard similarity between the critical KV index sets, $\mathcal{I}_t$ and $\mathcal{I}_{t+1}$, of adjacent decoding tokens within the same segment. The results provide clear evidence for our hypothesis. As shown in Figure 3(a) and (b), the similarity curves (bottom left) indicate that the Jaccard similarity remains consistently above 0.8 within *Current Segment* for both document QA and mathematical reasoning examples. This observation is also visually confirmed by the attention heatmaps (bottom right of Figure 3(a) and (b)). The bright, vertical bands in the heatmaps indicate that attention is persistently focused on the same regions of the prompt and previous output across the iterations of the current segment's generation. The observed temporal locality renders per-token KV retrieval unnecessary, motivating a semantic-aware strategy that amortizes the retrieval overhead across multiple tokens.

**Observation 2: Critical KVs exhibit distinct distribution patterns across the input and output sequences.** While temporal locality informs when to retrieve, the spatial distribution of critical KVs informs how to retrieve critical KV entries both efficiently and precisely.

As exemplified by the long-document QA task in Figure 3(a), we observe that critical KVs in the long input sequence are distributed sparsely. Specifically, the attention heatmap reveals that high-attention regions are sparsely dispersed across the long input sequence. This shows that the model must extract scattered information from distant parts of the context to generate the answer. In contrast, critical KVs in a long output sequence are often concentrated densely. As shown in Figure 3(b), for tasks like mathematical reasoning, the model's attention focuses densely on the previously generated segments, such as intermediate lemmas. As discussed in Section 3.1, existing KV retrieval methods typically employ fixed-size pages as the fundamental unit for management and selection. However, this approach ignores the distinct spatial distribution of critical KVs, leading to the retrieved pages containing numerous non-critical KV entries. This distinct distribution pattern poses a fundamental challenge for conventional page-level management mechanisms. This strongly motivates the design of a finer-grained KV management mechanism capable of adapting to this distinct distribution.

## 4 LOUISKV SYSTEM

### 4.1 SEMANTIC-AWARE ADAPTIVE RETRIEVAL

As established in Observation 1, the model tends to focus on a highly similar KV subset across multiple consecutive tokens during decoding, rendering per-token retrieval redundant and inefficient. To leverage this temporal locality, LouisKV introduces a semantic-aware adaptive retrieval strategy that operates at the granularity of *temporal segments*, thereby amortizing the retrieval cost over multiple decoding tokens.

The central challenge is to identify the boundaries of these segments accurately with low overhead. A naive approach is to compare the critical KV index sets of adjacent tokens. However, this approach incurs substantial computational costs, as the $Sel$ function itself involves dense matrix computations. LouisKV therefore employs a lightweight alternative (Xu et al., 2024; Liu et al., 2025). Specifically,

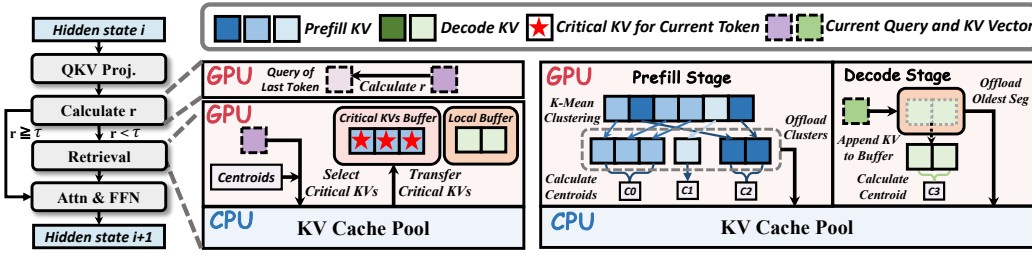

Figure 4: The design of LouisKV. (a) Retrieval is triggered when the query similarity $r$ drops below a threshold $\tau$, loading critical KV entries from the CPU cache pool. (b) During prefilling, K-means clustering is employed to group semantically similar KVs into clusters. During decoding, consecutively generated KVs are partitioned into temporal segments. These clusters and segments are then offloaded to a unified cache pool on the CPU. The detailed algorithm is provided in Appendix C.

for each transformer layer at each decoding step $t$, LouisKV computes the cosine similarity between the current query vector $q_t$ and the previous one $q_{t-1}$, averaged across all attention heads:

$$r_t = \frac{1}{H} \sum_{h=1}^{H} \mathrm{cosine}(q_{t-1}^h, q_t^h)$$

where $H$ is the total number of attention heads and $q_t^h$ is the query vector for head $h$ at step $t$. As shown in Figure 4(a), when this similarity score $r$ drops below a predefined threshold $\tau$, LouisKV identifies a semantic boundary and triggers a KV retrieval operation. This operation selects the critical units that scored the highest by computing the similarity between the current query $q_t$ and the centroids $C$ of all units, and subsequently loads the corresponding KV cache from CPU to GPU.

## 4.2 DECOUPLED FINE-GRAINED MANAGEMENT

After determining when to trigger the retrieval, the subsequent challenge is how to manage the KV cache to improve retrieval precision. According to Observation 2, traditional coarse-grained page-level management failed to efficiently adapt to the distinct distribution patterns across the input and output sequences. LouisKV therefore designs a decoupled fine-grained management scheme (Figure 4(b)) that tailors retrieval units separately for each phase.

**Prefill Stage.** To manage the sparse critical KVs in the input sequence, LouisKV introduces a cluster-granularity strategy. Since the attention score for a query vector $q_t$ is primarily determined by its dot product with each key vector, key vectors that are similar in semantic space tend to exhibit similar attention scores (Liu et al., 2024b). Accordingly, LouisKV employs the k-means clustering algorithm to group similar key vectors into the same cluster. Subsequently, LouisKV calculates the centroid $C_i$ for each cluster by averaging all its key vectors. Unlike traditional page-level strategies, the key vectors within the same cluster are semantically similar, allowing this centroid to serve as a more precise index for subsequent retrieval. This KV clustering and offloading process occurs asynchronously with the prefill forward pass, avoiding blocking the forward computation.

**Decode Stage.** To handle the highly dense attention pattern in the output sequence, LouisKV leverages the semantic boundaries identified in Section 4.1 to partition the generated KV cache into multiple temporal segments. Similar to the semantic clusters from the prefill stage, each temporal segment is represented by an index vector derived from averaging its key vectors. To prevent excessive GPU memory consumption during decoding, LouisKV maintains a fixed-size local buffer on the GPU to cache the most recently generated KV segments. When this buffer becomes full, the oldest segment will be offloaded to CPU memory. These offloaded temporal segments from the decode stage, together with the semantic clusters from the prefill stage, form a unified KV cache pool in CPU memory.

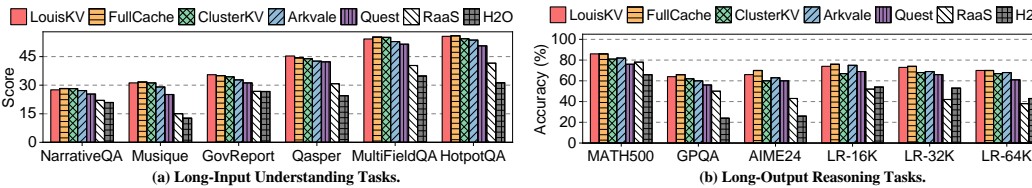

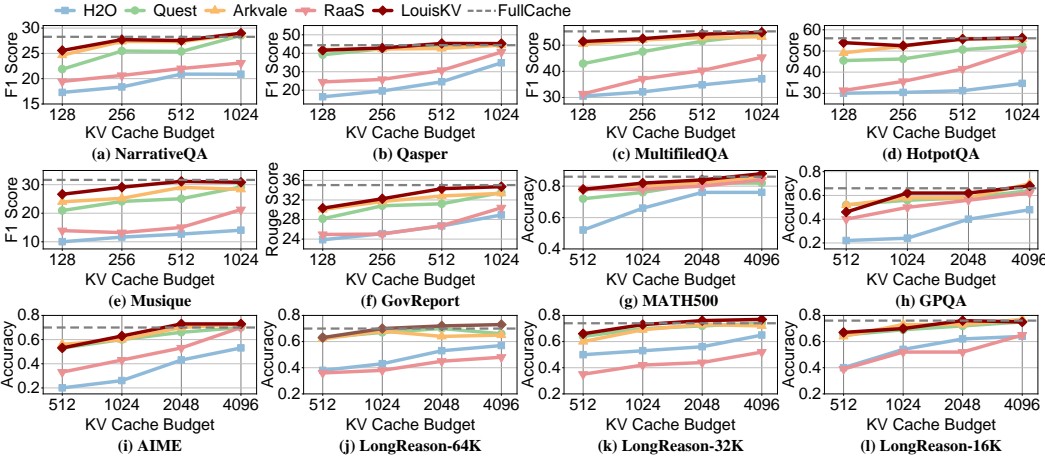

Figure 5: Performance comparison of LouisKV against six baseline methods under the fixed KV cache budget. Subplot (a) presents the results on six long-input understanding tasks using Llama-3.1-8B-Instruct. Subplot (b) presents the results on six long-output reasoning tasks using Qwen3-8B.

Figure 6: Performance comparison across various cache budgets. Subplots (a-f) present the results on six long-input understanding tasks. Subplots (g-l) present the results on six long-output reasoning tasks. For detailed experimental results on more models, please refer to Appendix D.1.

## 4.3 SYSTEM OPTIMIZATIONS

To enhance overall system efficiency, LouisKV incorporates several system-level and kernel-level optimizations. First, we implement a custom Triton (Tillet et al., 2019) kernel to efficiently execute the clustering operation during prefilling. Second, we introduce a group-consistent selection strategy that selects a unified set of KV clusters and segments for an entire query group, which efficiently adapts to the Grouped-Query Attention (Ainslie et al., 2023) by avoiding redundant data transfers (see Appendix B for details). Finally, since these clusters or segments contain a variable number of KV entries, selecting critical items under a fixed budget in standard frameworks like PyTorch is complex and inefficient. To address this, we design a highly optimized CUDA kernel that supports rapid, batched selection of critical KVs under a given budget. Furthermore, to facilitate efficient transfer of selected KV entries between CPU and GPU, we use the DGL library (Wang, 2019) to directly transfer specific rows from CPU tensors to the GPU device, rather than first gathering the rows on the CPU and then transferring them.

## 5 EVALUATION

### 5.1 EXPERIMENTAL SETUP

**Datasets and Models.** To comprehensively evaluate LouisKV, we select the following datasets: (1) *Long-Input, Short-Output.* This type of task primarily involves document understanding. Thus, we select six representative and popular datasets from the LongBench (Bai et al., 2023) benchmark.[1] To further stress-test our system's performance on ultra-long sequences, we employ the RULER (Hsieh et al., 2024) benchmark with context lengths ranging from 32K to 128K. (2) *Short-Input, Long-Output.* We chose three challenging mathematical and scientific reasoning datasets, sampling 50 problems from each: MATH500 (Hendrycks et al., 2021) and AIME (Jia, 2025), which require

---

[1]The selected datasets are NarrativeQA, Qasper, MultiFieldQA, HotpotQA, Musique, and GovReport.

multi-step mathematical problem-solving, and GPQA (Rein et al., 2024), which comprises graduate-level questions demanding complex reasoning. (3) *Long-Input, Long-Output.* We choose the LongReason (Ling et al., 2025) dataset. This dataset synthetically extends short-context reasoning problems into complex versions requiring long-range dependency, and we sample 100 problems from each of its 16K, 32K, and 64K input-length subsets, denoted as LongReason-16K, LongReason-32K, and LongReason-64K, respectively. Our evaluation is conducted on two models: Qwen3-8B (Yang et al., 2025) and Llama-3.1-8B-Instruct (Grattafiori et al., 2024). We leverage the thinking mode of Qwen3-8B to handle long-output tasks requiring complex reasoning, while Llama-3.1-8B-Instruct addresses traditional long-input, short-output tasks.

**Baselines.** We compare LouisKV against state-of-the-art KV cache optimization methods, including KV dropping methods such as H2O (Zhang et al., 2023) and RaaS (Hu et al., 2025), KV retrieval methods such as Quest (Tang et al., 2024), Arkvale (Chen et al., 2024a) and ClusterKV (Liu et al., 2024b). We set a uniform KV cache budget $B$ for all methods. Following the setting of Quest, we preserve the complete KV cache for the first two layers and retain the sink tokens $S$ and a local buffer $W$ on the GPU across all methods, where the values of $S$ and $W$ are adjusted according to the specific task. To maintain a fair comparison, we adapt the implementations of RaaS, Quest, and Arkvale to be group-consistent. For method-specific parameters, the page size is set to 16 for Quest, Arkvale, and RaaS while the average cluster size is set to 16 for LouisKV and ClusterKV on the input sequence. Our proposed LouisKV employs varying similarity thresholds $\tau$ for the output sequence, adapted to the specific model. We determine its optimal value via a one-time calibration process (detailed in Section 5.4) to ensure the best trade-off between efficiency and accuracy.

**Environment.** Our server is equipped with a single NVIDIA A6000 GPU (48 GB) and an AMD Ryzen 9 5950X 16-Core Processor. The CPU and GPU are interconnected via PCIe 4.0 ×16 (32GB/s). All experiments run on Ubuntu 20.04 with Linux kernel 5.4.0 and CUDA 12.4.

## 5.2 ACCURACY EVALUATION

**Long-Input Understanding Tasks.** We set the system parameters to $S = 32$, $W = 512$, and $\tau = 0.85$. For a comprehensive analysis, Figure 5(a) details the performance on individual datasets under a specific budget of 512, whereas Figure 6(a-f) presents the score across varying budgets. The results show that LouisKV consistently surpasses other page-level retrieval methods and dropping methods across nearly all long-input tasks. Specifically, with a budget of 512, LouisKV improves the average score by 1.1% over Arkvale and 3.3% over Quest. Although LouisKV is marginally lower than the cluster-level method, ClusterKV (a difference of approximately 0.4%), this slight gap represents a strategic trade-off in our system design. While both methods employ clustering for the input sequences, LouisKV distinguishes itself by incorporating a segment-aware retrieval mechanism that triggers retrieval only at semantic boundaries. This design choice yields a significant enhancement in inference efficiency for a negligible cost in accuracy. In summary, the superior performance of LouisKV validates the effectiveness of our clustering management designed for the input sequence. Unlike coarse-grained page-level management, our approach more precisely identifies and retains the most critical KVs, leading to higher accuracy under limited budgets.

**Long-Output Reasoning Tasks.** For long-output tasks that demand complex reasoning, we apply different parameter configurations. In short-input, long-output tasks, we set $S = 500$, $W = 128$, and $\tau = 0.7$. In long-input, long-output tasks, we set $S = 64$, $W = 256$, and $\tau = 0.7$. The maximum generation length is 16K tokens, extended to 32K for the AIME benchmark. Figure 6(g-l) show that LouisKV achieves the best or second-best accuracy across various budgets. Meanwhile, Figure 5(b) highlights the performance on individual benchmarks, revealing that KV dropping methods suffer from severe accuracy degradation, particularly on challenging tasks like AIME and LongReason. Under a budget of 1024, LouisKV surpasses the page-level methods Arkvale and Quest by 2.0% and 4.8% in average accuracy, respectively, and outperforms the cluster-level method ClusterKV by 3.8%. These results conclusively validate the effectiveness of our decoupled KV management scheme, demonstrating its capability to precisely preserve critical information across diverse input and output distributions.

**Robustness on RULER Benchmark.** To evaluate LouisKV's robustness in extreme long-context scenarios, we evaluate performance across 13 datasets from the RULER suite. As shown in Table 4 of the Appendix D.1, LouisKV demonstrates strong performance from 32K to 128K context lengths

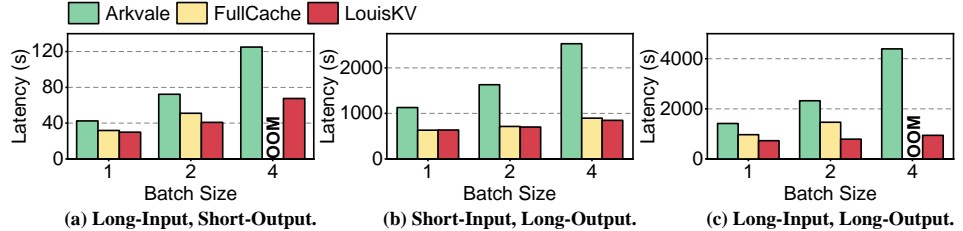

Figure 7: End-to-end latency comparison under different batch sizes and long-sequence scenarios.

| Input Length | 2K | 4K | 8K | 16K | 32K | 64K |
|---|---|---|---|---|---|---|
| **FullCache TTFT (s)** | 0.36 | 0.72 | 1.51 | 3.40 | 8.61 | 22.81 |
| **LouisKV TTFT (s)** | 0.37 | 0.76 | 1.60 | 3.52 | 8.85 | 23.49 |
| **K-Means Latency (s)** | 0.06 | 0.11 | 0.16 | 0.27 | 0.78 | 2.34 |

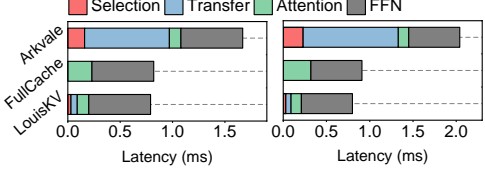

Table 1: Comparison of TTFT between FullCache and LouisKV, and the standalone K-Means latency across different input lengths.

Figure 8: Per-layer latency breakdown during the decoding phase. Left: short-input, long-output. Right: long-input, long-output.

while setting the KV budget to 2K tokens. At 32K and 64K, its accuracy is comparable to the lossless FullCache baseline and surpasses Arkvale by 2.6% and 4.7%, respectively. At 128K, FullCache fails with an OOM error on our hardware, highlighting the fundamental necessity of retrieval-based methods for such extreme contexts. In contrast, LouisKV completes the tasks successfully and consistently outperforms Arkvale across all tested lengths. This performance gap underscores the effectiveness of our design in accurately identifying and retrieving the most critical KV entries, even in ultra-long sequences.

## 5.3 EFFICIENCY EVALUATION

We evaluate the inference efficiency of LouisKV in comparison with FullCache and Arkvale. Specifically, the FullCache baseline, which retains the entire KV cache on GPU, is implemented using FlashAttention-2 (Dao, 2023). The evaluation covers three long-sequence patterns: long-input, short-output (32K input, 512 output); short-input, long-output (500 input, 16K output); and long-input, long-output (32K input, 16K output). Following the experimental setup from the Section 5.2, we use Llama-3.1-8B-Instruct for long-input scenarios and Qwen3-8B for long-output scenarios.

**End-to-End Latency.** As illustrated in Figure 7, compared to Arkvale, LouisKV achieves up to $1.9\times$, $2.9\times$ and $4.7\times$ speedups across the three patterns, respectively. This efficiency gain stems from our semantic-aware retrieval strategy, which significantly reduces computation and data transfer overhead, and highly optimized kernels that enhance the efficiency of importance estimation and data movement. Specifically, by triggering retrieval only at semantic segment boundaries, LouisKV drastically reduces the retrieval frequency (detailed analyses regarding segment lengths and per-layer boundaries are provided in Appendix D.5 and D.6). This architectural advantage becomes more pronounced as the batch size increases, because larger batches exacerbate the data transfer bottleneck. LouisKV exhibits comparable latency compared with FullCache. When processing long sequences with larger batches, FullCache encounters OOM errors due to its significant memory footprint. In contrast, LouisKV offloads the entire KV cache to CPU memory, enabling support for larger batch sizes, thereby achieving higher system throughput.

**Prefill Latency.** We evaluate the Time to First Token (TTFT) of LouisKV compared to FullCache across input lengths ranging from 2K to 64K. As detailed in Table 1, LouisKV exhibits performance comparable to FullCache, introducing only a marginal average latency increase of approximately 3.5%. This negligible overhead attributes to our Triton-optimized K-means kernel and the asynchronous execution pipeline. To quantify this overhead, we measure the standalone latency of the clustering process. While the clustering cost scales with input length, the asynchronous pipeline successfully overlaps this computation with the inference process. Consequently, LouisKV maintains high prefill efficiency without imposing a significant bottleneck.

Figure 9: Ablation studies. (a) Accuracy comparison of our adaptive stride retrieval against a fixed stride baseline. (b) Accuracy comparison of our decoupled, fine-grained management against a coarse-grained baseline. (c) Impact of the similarity threshold $\tau$ on accuracy and latency. (d) Efficiency contribution of individual system optimizations.

**Decode Latency Breakdown.** We evaluate the latency breakdown for a single transformer layer during decoding. As shown in Figure 8, although Arkvale retrieves only a subset of the KV cache, the overhead associated with retrieval, including both selection and transfer, still accounts for 65% of the total latency, highlighting the necessity to reduce this overhead. LouisKV drastically reduces this overhead, lowering its contribution to just 11% of the total latency. We also provide the analyses of memory efficiency in Appendix D.2.

## 5.4 ABLATION STUDY

**Effectiveness of Core Strategies.** We independently validate the effectiveness of the two core strategies in LouisKV. First, we evaluate the adaptive retrieval against a baseline that employs a fixed-stride retrieval. The results in Figure 9(a) demonstrate that our adaptive approach consistently achieves higher accuracy across all tasks, with gains of up to 8% in long-output reasoning tasks. This indicates that dynamically triggering retrieval at semantic boundaries captures critical dependencies more effectively than rigid, fixed-stride retrieval. Furthermore, we assess the effectiveness of our decoupled, fine-grained management scheme by comparing it to a coarse-grained, page-level baseline. To strictly isolate the impact of memory management granularity from the retrieval policy, we configure both methods to use an identical per-token retrieval setting. As shown in Figure 9(b), our management scheme consistently outperforms the page-level baseline, achieving improvements of up to 6.3%. A detailed analysis of this ablation study is provided in Appendix D.3.

**Impact of Similarity Threshold $\tau$.** The similarity threshold $\tau$ serves as a key hyperparameter that governs the trade-off between accuracy and efficiency. As shown in Figure 9(c), a higher $\tau$ leads to more frequent retrievals, thus improving inference accuracy. However, this comes at the cost of higher latency due to the overhead of retrieval. The results show that for Qwen3-8B, $\tau = 0.7$ represents an ideal balance, achieving significant efficiency gains while maximally preserving inference accuracy. Notably, we calibrate $\tau$ as a one-time, model-specific parameter using a small validation set (e.g. 300 examples). Once determined, this optimal value generalizes to all long-context tasks, eliminating the need for per-task tuning. Further details are in Appendix D.4.

**Impact of System Optimizations.** Finally, we quantify the efficiency contribution of the system optimizations in LouisKV, testing on the Qwen3-8B model in long-input, long-output scenarios. As shown in Figure 9(d), we incrementally apply each optimization on top of a baseline system. Semantic-Aware Retrieval (SR) contributes the largest performance improvement, achieving a nearly $2.6\times$ speedup by drastically reducing the overhead from redundant retrieval and data loading operations. Building on this, Group-Consistent Selection (GS) and our Custom Retrieval Kernel (CK) provide additional performance gains of 13.1% and 15.7%, respectively.

## 6 CONCLUSION

In this paper, we propose LouisKV, an effective KV retrieval framework designed for various long-sequence scenarios. Our work is motivated by two key observations: the strong temporal locality of critical KVs, and their distinct distribution patterns across input and output sequences. Building on these insights, LouisKV introduces a semantic-aware adaptive retrieval strategy and a decoupled management scheme. Evaluation results show that LouisKV achieves up to a $4.7 \times$ speedup compared to state-of-the-art KV retrieval methods while maintaining near-lossless inference accuracy.

## ACKNOWLEDGMENTS

This work is mainly supported by the National Key Research and Development Program of China under Grant No. 2023YFB4502702, the Natural Science Foundation of China under Grant No. 62472007, 62332021 and 62306056.

## ETHICS STATEMENT

Our research is conducted in full compliance with the ICLR Code of Ethics. We are committed to upholding the highest standards of academic integrity and have addressed key ethical considerations as follows:

- **Research Integrity:** We have upheld research integrity by ensuring all data, methods, and results are presented truthfully and without misrepresentation. Our methodology is described with sufficient detail to support independent verification and reproducibility.
- **Data and Privacy:** The datasets utilized in our experiments are publicly available and were handled in strict accordance with their respective licenses and usage terms. Our study did not involve human subjects or personally identifiable information.
- **Attribution and Contribution:** Proper attribution has been given to all prior work through comprehensive citations. All authors listed have made significant intellectual contributions to this research and have collectively approved this submission.
- **Societal Impact:** We have considered the potential societal impacts of our work. While the intended applications are beneficial, we acknowledge the possibility of misuse and advocate for the responsible development and deployment of our findings.

## REPRODUCIBILITY STATEMENT

To ensure the reproducibility of our results, we have provided detailed descriptions of our experimental setup, hyperparameters, and implementation details. We plan to make the source code publicly available upon publication to facilitate verification and future research.

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

# A    DETAILED BOTTLENECK ANALYSIS OF KV RETRIEVAL METHODS

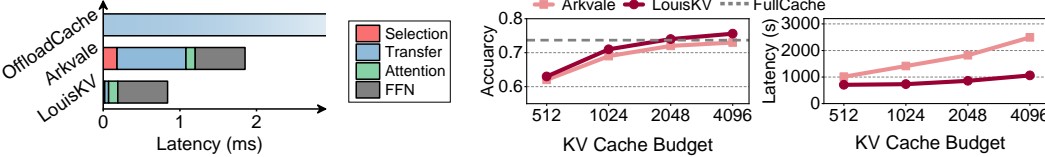

Figure 10: Bottleneck analysis of KV retrieval methods. Left: Per-layer latency breakdown comparing LouisKV with other methods. OffloadCache serves as a baseline that preloads the full KV cache of the next layer from CPU during inference, while Arkvale is a representative state-of-the-art retrieval method. Right: Inference accuracy (middle) and end-to-end latency (right) of LouisKV and Arkvale in long-output scenarios under different KV cache budgets.

While existing KV retrieval methods avoid GPU Out-Of-Memory errors by offloading the entire KV cache to CPU memory, they still face two fundamental bottlenecks regarding inference efficiency and retrieval accuracy.

First, the per-token retrieval strategy incurs prohibitive cumulative overhead. As illustrated in Figure 10 (left), although retrieval methods like Arkvale attempt to reduce KV cache transfer costs by loading less data, they introduce an additional computational overhead to select the critical KV cache pages. Meanwhile, due to the limited CPU-GPU bandwidth, the data transfer overhead continues to be a primary performance bottleneck. While this overhead might seem minor for a single token generation, it accumulates linearly as each decoding token in a long-output sequence requires a new retrieval operation. This cumulative effect becomes substantial in long-output scenarios, leading to a dramatic increase in total inference latency, as shown in Figure 10 (right) for Arkvale.

Second, the coarse-grained page-level management scheme leads to suboptimal retrieval performance, creating a dilemma between efficiency and accuracy. These methods partition the sequence into fixed-size pages and use the page as the minimum unit for retrieval, which introduces a difficult trade-off. On the one hand, it results in a significant accuracy drop under tight KV cache budgets, as shown in Figure 10 (middle). On the other hand, allocating a larger budget to prevent this accuracy loss results in significantly higher end-to-end latency, as depicted in Figure 10 (right). In contrast, by retrieving the critical KVs precisely, LouisKV maintains high inference accuracy even under a small budget while simultaneously achieving high efficiency.

# B    DETAILS OF SYSTEM OPTIMIZATION

**Group-consistent Selection.** Existing retrieval methods are primarily designed for Multi-Head Attention architectures, where the number of query heads $h_q$ equals the number of KV heads $h_{kv}$. However, modern models often adopt Grouped-Query Attention, in which a group of $g = h_q/h_{kv}$ query heads share the same KV head. To avoid excessive transfer overhead caused by retrieving distinct KV heads for each query head, we propose a group-consistent retrieval strategy. This approach selects a common set of KV indices for all queries within a group, guided by a group-aggregated score $A_t^i$, computed by averaging the softmax-normalized attention scores from each query:

$$A_t^i = \frac{1}{g} \sum_{j=1}^{g} \text{softmax} \left( \frac{q_t^{i*g+j}(C^i)^T}{\sqrt{d}} \right)$$

where $q_t^{i*g+j}$ is the query of the $j$-th head within group $i$ at step $t$, and $C^i$ is the matrix containing the centroid vectors of all clusters and segments belonging to the $i$-th KV head, the one shared by the $i$-th query group. Based on this aggregated score, LouisKV ranks all KV clusters and segments and selects the top-scoring units. This strategy ensures that all requests from the group are unified into a single, coalesced data transfer, perfectly aligning with the computational pattern of the GQA architecture and fundamentally eliminating the transfer overhead. Notably, for conventional MHA architectures where the group size $g = 1$, our group-aware strategy naturally simplifies to a standard per-head retrieval mechanism, thus providing a unified approach for different attention structures.

## C  ALGORITHM OVERVIEW

---

**Algorithm 1** The LouisKV Algorithm

---

1: **Input:** Model $M$, Input Sequence $X_{inp}$, KV Cache Budget $B$, Similarity Threshold $r$, A KV Manager $kvm$
2: **Output:** Generated Token Sequence $O$

3: **function** KVM.STORE_CACHE($K, V$, stage)
4:     **if** stage == 'prefill' **then**
5:         $K_{clusters} \leftarrow$ KMeans($K$)
6:         **for each** cluster $c$ in $K_{clusters}$ **do**
7:             $centroid \leftarrow$ ComputeCentroid($c$);
8:             Store $centroid$ on GPU;
9:         **end for**
10:        Offload $(K, V)$ to CPU memory pool asynchronously
11:    **else if** stage == 'decode' **then**
12:        $KV_{local} \leftarrow [KV_{local} : (k_t, v_t)]$
13:        **if** $KV_{local}$ is full **then**
14:            $K_{seg}, V_{seg} \leftarrow$ the oldest KV segment in $KV_{local}$
15:            $centroid \leftarrow$ ComputeCentroid($K_{seg}$);
16:            Store $centroid$ on GPU
17:            Offload $(K_{seg}, V_{seg})$ to CPU memory pool asynchronously.
18:        **end if**
19:    **end if**
20: **end function**

21: **function** KVM.RETRIEVE($q_t, B$)
22:     $Scores \leftarrow$ Compute similarity scores between query $q_t$ and all centroids
23:     $TopIndices \leftarrow$ Select the set of top-scoring clusters and segments based on $Scores$ and budget $B$
24:     $KV_{critical} \leftarrow$ Load the corresponding KVs of $TopIndices$ from CPU memory pool
25:     **return** $KV_{critical}$
26: **end function**

27: **procedure** LOUISKV($M, X_{inp}, B, r, kvm$)

28:     // Phase 1: Prefill Stage
29:     $K_{inp}, V_{inp} \leftarrow$ Process $X_{inp}$ with model $M$ to get initial KV cache
30:     $kvm$.store_cache($K_{inp}, V_{inp}$, 'prefill')

31:     $O \leftarrow \emptyset$; $q_{prev} \leftarrow$ None;
32:     // Phase 2: Decode Stage
33:     **for** $t = 1, 2, \ldots, T$ **do**
34:         $q_t, k_t, v_t \leftarrow$ Generate query and KV cache from the previous token's state
35:         is_new_segment $\leftarrow (t == 1) \vee$ (CosineSimilarity($q_t, q_{prev}$) $< r$); $q_{prev} \leftarrow q_t$
36:         **if** is_new_segment == True **then**
37:             $KV_{critical} \leftarrow kvm$.retrieve( $q_t, B$ )
38:             $kvm$.store_cache($k_t, v_t$, 'decode'); $KV_{attn} \leftarrow [KV_{critical} : KV_{local}]$
39:         **else if** is_new_segment == False **then**
40:             $kvm$.store_cache($k_t, v_t$, 'decode'); $KV_{attn} \leftarrow [KV_{critical} : KV_{local}]$
41:         **end if**
42:         $o_t \leftarrow$ Model $M$ Generate token using $q_t$ and $KV_{attn}$; $O$.append($o_t$);
43:     **end for**
44:     **return** $O$
45: **end procedure**

---

# D    DETAILED EXPERIMENTAL RESULTS AND ANALYSIS

## D.1    ACCURACY EVALUATION

| Methods | Long-Input, Short-Output Tasks | | | | | |
|---|---|---|---|---|---|---|
| | NarrativeQA | Qasper | MultifieldQA | HotpotQA | Musique | GovReport |
| *Llama3.1-8B-Instruct* | 28.26 | 44.38 | 55.27 | 55.97 | 31.68 | 35.02 |
| H2O | 20.90 | 24.50 | 34.82 | 31.29 | 12.70 | 26.69 |
| RaaS | 22.01 | 30.72 | 40.31 | 41.54 | 15.02 | 26.74 |
| Quest | 25.35 | 43.65 | 51.49 | 51.63 | 25.09 | 32.20 |
| Arkvale | 27.32 | 42.59 | 52.88 | **55.92** | 29.07 | 32.79 |
| LouisKV | **27.53** | **45.31** | **54.22** | 55.64 | **31.20** | **33.30** |
| *Qwen2.5-7B-Instruct* | 20.1 | 39.29 | 46.66 | 55.84 | 25.88 | 31.4 |
| H2O | 7.36 | 20.04 | 31.07 | 30.75 | 15.29 | 22.48 |
| RaaS | 10.7 | 25.21 | 35.90 | 39.43 | 15.83 | 26.2 |
| Quest | 15.41 | 34.56 | 44.75 | 45.58 | 22.51 | 29.21 |
| Arkvale | 17.64 | **38.78** | 44.40 | **53.16** | 25.81 | **30.6** |
| LouisKV | **19.39** | 37.67 | **46.63** | 52.14 | **28.51** | 30.57 |

Table 2: Accuracy comparison on Long-Input Understanding tasks with a KV cache budget of 512 tokens across all methods. The best results for each model family (excluding the baseline) are in **bold**.

| Methods | Short-Input, Long-Output Tasks | | | Long-Input, Long-Output Tasks | | |
|---|---|---|---|---|---|---|
| | MATH500 | GPQA | AIME | LongReason-64K | LongReason-32K | LongReason-16K |
| *Qwen3-8B* | 0.86 | 0.66 | 0.70 | 0.70 | 0.74 | 0.76 |
| H2O | 0.66 | 0.24 | 0.26 | 0.43 | 0.53 | 0.54 |
| RaaS | 0.78 | 0.50 | 0.43 | 0.38 | 0.42 | 0.52 |
| Quest | 0.76 | 0.56 | 0.60 | 0.67 | 0.70 | 0.73 |
| Arkvale | 0.82 | 0.60 | 0.63 | 0.68 | 0.68 | **0.75** |
| LouisKV | **0.86** | **0.62** | **0.66** | **0.70** | **0.73** | 0.74 |
| *DeepSeek-R1-Distill-Qwen-7B* | 0.84 | 0.32 | 0.50 | 0.20 | 0.37 | 0.40 |
| H2O | 0.74 | 0.26 | 0.23 | 0.15 | 0.18 | 0.28 |
| RaaS | 0.82 | 0.30 | 0.40 | 0.17 | 0.23 | 0.32 |
| Quest | **0.86** | 0.38 | 0.43 | 0.16 | 0.34 | 0.34 |
| Arkvale | 0.84 | 0.44 | **0.53** | 0.20 | 0.38 | 0.38 |
| LouisKV | **0.86** | **0.48** | 0.50 | **0.22** | **0.39** | **0.42** |

Table 3: Accuracy comparison on Long-Output Reasoning tasks with a KV cache budget of 1024 tokens across all methods. The best results for each model family (excluding the baseline) are in **bold**.

| Methods | S1 | S2 | S3 | MK1 | MK2 | MK3 | MV | MQ | FWE | CWE | QA1 | QA2 | VT |
|---|---|---|---|---|---|---|---|---|---|---|---|---|---|
| *FullCache (32K)* | 100 | 100 | 100 | 100 | 100 | 100 | 97.12 | 99.75 | 95.33 | 52.00 | 74.5 | 54.0 | 99.8 |
| Arkvale | **100** | 99.5 | 99.5 | 99.5 | 99.5 | 92 | 94.12 | **99.75** | 92.17 | 31.30 | 73.5 | 54.0 | 99.2 |
| LouisKV | **100** | **100** | **100** | **100** | **100** | **99.5** | **96.12** | **99.75** | **92.67** | **49.85** | **75.0** | **55.5** | **99.5** |
| *FullCache (64K)* | 100 | 100 | 100 | 100 | 99 | 95.5 | 97.62 | 99.75 | 86.17 | 12.05 | 72.0 | 49.0 | 98.6 |
| Arkvale | **100** | 98.5 | 99 | 99.5 | 96.5 | 57.5 | 94.50 | 99.75 | 78.17 | 2.35 | 71.5 | 49.0 | 95.2 |
| LouisKV | **100** | **100** | **100** | **100** | **98.5** | **92** | **96.88** | **100** | **85.33** | **11.80** | **72.0** | **50.0** | **96.5** |
| *FullCache (128K)* | OOM | OOM | OOM | OOM | OOM | OOM | OOM | OOM | OOM | OOM | OOM | OOM | OOM |
| Arkvale | **100** | 99.5 | 97.5 | 96 | 73 | 6.5 | 72.00 | 90.62 | 60.17 | 0.00 | 68.0 | **45.0** | 79.5 |
| LouisKV | **100** | **100** | **100** | **98** | **79** | **25** | **90.50** | **96.62** | **62.33** | **0.10** | **69.0** | 44.0 | **80.7** |

Table 4: Accuracy on RULER across context lengths from 32K to 128K. In the header, S1-S3, MK1-MK3, MV, and MQ represent the niah_single_1-3, niah_multikey_1-3, niah_multivalue, and niah_multiquery tasks, respectively.

| Method | Memory | Parameters |
|--------|--------|------------|
| FullCache | $4Lhd_h(n+m)$ | - |
| Quest | $4Lhd_h(n+m+\frac{n+m}{p})$ | page size: $p$ |
| Arkvale | $4Lhd_h(B+\frac{n+m}{p})$ | page size: $p$, budget size: $B$ |
| LouisKV | $4Lhd_h(B+\frac{n}{2c}+\frac{m}{2s})$ | cluster size: c, segment size: $s$, budget size: $B$ |

Table 5: Theoretical analysis of GPU KV cache memory footprint for different methods, where $n$ is the input length and $m$ is the output length.

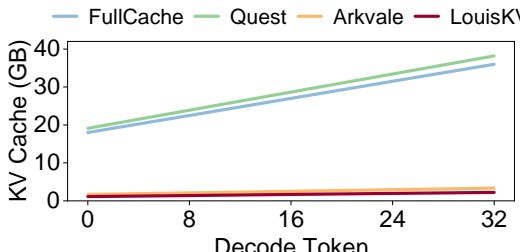

Figure 11: Comparison of KV cache memory consumption during the decoding phase for different methods, with a prefill length of 32K tokens and a batch size of 4.

## D.2 MEMORY FOOTPRINT ANALYSIS

In this section, we analyze the KV cache memory footprint of different methods. We assume the number of layers is $L$, the number of heads is $h$, the head dimension is $d_h$, the input sequence length is $n$, and the output sequence length is $m$. All KV cache is stored in FP16, leading to a base factor of $4Lhd_h$. As detailed in Table 5, FullCache retains the entire KV cache on the GPU. Quest builds upon this by adding indexing overhead for each page of size $p$. Both Arkvale and LouisKV are offloading methods that keep a budget of size $B$ on the GPU, but they differ in their indexing overhead. Specifically, both Quest and Arkvale create two index vectors for each key page, whereas LouisKV requires only one for each semantic cluster or segment. We empirically validate this analysis by comparing memory consumption in a long-input scenario (32K prefill tokens), with results shown in Figure 11. The memory usage of FullCache and Quest grows linearly with decoded tokens, making them susceptible to out-of-memory errors. In contrast, Arkvale and LouisKV maintain consistently low memory footprints. As future work, LouisKV's memory consumption could be further reduced by offloading KV cache indices to CPU DRAM.

## D.3 EFFECTIVENESS OF CORE STRATEGIES

In this section, we conduct ablation studies to independently analyze the effectiveness of LouisKV's two core designs, *adaptive semantic-aware retrieval strategy* and *decoupled fine-grained management scheme*, to increase inference accuracy. We follow the parameter settings in Section 5, testing on long-input understanding tasks with the Llama3.1-8B-Instruct model and on long-output reasoning tasks with the Qwen3-8B model.

**Fixed Stride vs. Adaptive Stride.** To validate the superiority of our adaptive retrieval strategy, we compare it against a baseline that employs a fixed-stride retrieval. For a fair comparison, the baseline triggers a retrieval every 5 tokens for long-input tasks and every 16 tokens for long-output tasks. These values are chosen to closely match the average retrieval frequency of our adaptive strategy on the respective tasks. As shown in Figure 12(a), our adaptive retrieval strategy achieves higher accuracy across the majority of tasks. This result strongly indicates that a fixed-stride retrieval method can degrade accuracy by failing to retrieve information at critical points. In contrast, our adaptive strategy more precisely identifies the critical decoding points where information must be updated, leading to better inference accuracy.

**Coarse-grained vs. Fine-grained Management.** Next, we evaluate the contribution of our decoupled, fine-grained management mechanism, comparing it against a baseline that applies a uniform

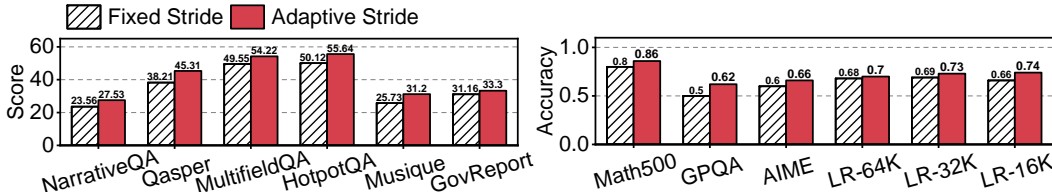

Figure 12: Ablation study on retrieval strategy. We compare the inference accuracy of our adaptive semantic-aware retrieval against a fixed-stride retrieval baseline. The adaptive strategy achieves higher accuracy across most tasks by retrieving information at critical decoding points.

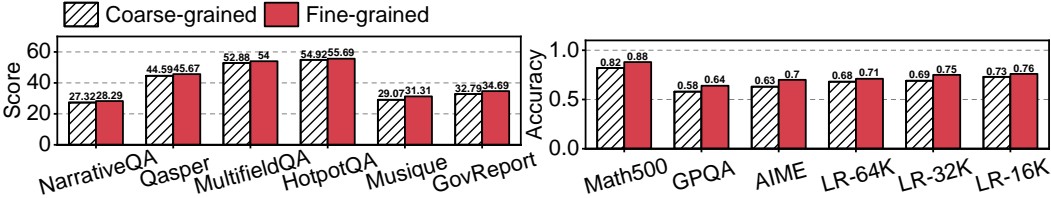

Figure 13: Ablation study on management scheme. We compare the inference accuracy of our decoupled, fine-grained management against a coarse-grained page-level baseline. The superior performance of our fine-grained approach validates the necessity of its tailored design: semantic clusters for inputs and temporal segments for outputs.

page-level management strategy. To ensure a fair comparison, we force both our strategy and the baseline to perform a retrieval at every decoding step. We set the page size of baseline to 16, aligning it with the average cluster and segment size used in our approach. As depicted in Figure 13(b), our fine-grained management strategy consistently outperforms the coarse-grained baseline. This result confirms the necessity of our tailored design: semantic clusters are better suited for the sparse attention patterns of input sequences, while temporal segments effectively capture the dense attention patterns of output sequences. This tailored design is crucial for achieving high inference accuracy.

### D.4 IMPACT OF SIMILARITY THRESHOLD $\tau$.

The threshold $\tau$ is a key hyperparameter that governs the trade-off between accuracy and efficiency. It defines the sensitivity for triggering a new KV retrieval operation. A lower value of $\tau$ imposes a stricter criterion for identifying a semantic boundary, thereby reducing the frequency of retrievals. While this enhances inference efficiency, it typically leads to accuracy degradation by potentially failing to update the set of critical KVs. Conversely, a higher $\tau$ results in better inference accuracy. However, the frequent retrievals incur additional computational and data transfer overhead, thus reducing overall efficiency. We experimentally analyze the impact of $\tau$ on both inference accuracy and efficiency.

**Analysis of Inference Accuracy.** To quantify the impact of $\tau$, we conducted a series of ablation studies on long-input and long-output tasks using the Llama-3.1-8B and Qwen3-8B models, respectively. As shown in Figure 14, a consistent trend is observed: model accuracy improves on all tasks with an increasing $\tau$. This is because a higher $\tau$ leads to more frequent retrievals, ensuring that critical information is recalled promptly. More importantly, long-input tasks (Figure 14(a-c)) exhibit significantly higher sensitivity to the value $\tau$ than long-output reasoning tasks (Figure 14(d-f)). We attribute this to the difference in task nature: to generate an answer, long-input understanding tasks often require shifting attention between disparate parts of the context, thus demanding a more responsive tan to capture these semantic changes. In contrast, long-output reasoning tasks exhibit stronger local coherence, as each reasoning step primarily attends to consistent context. This results in naturally longer semantic segments, making the model's performance less sensitive to the precise value of tan.

**Analysis of Inference Efficiency.** To quantify the impact of $\tau$ on efficiency, we measured the retrieval frequency and inference latency across different thresholds. As shown in Figure 14(g-

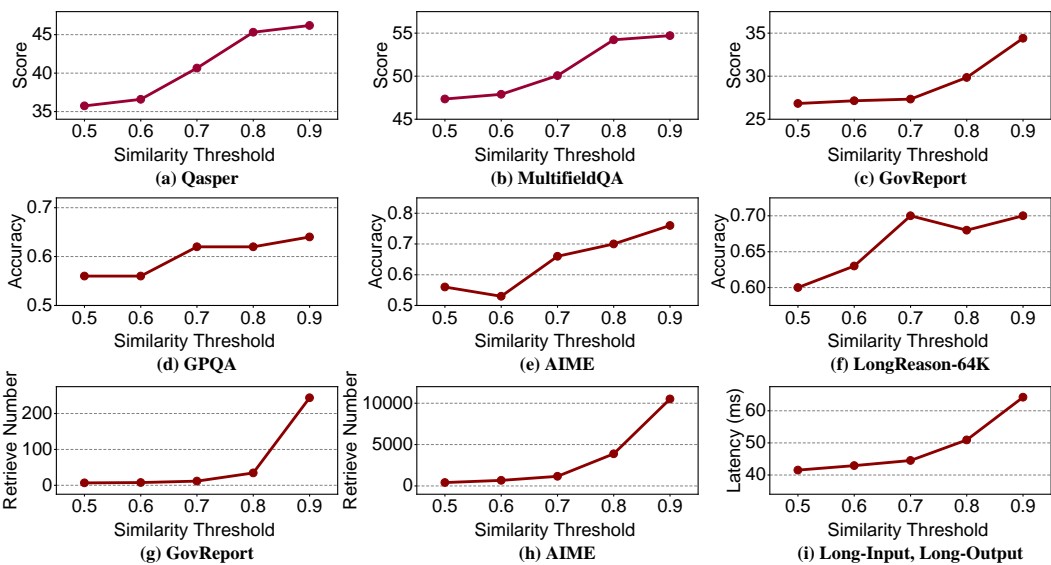

Figure 14: Impact of the similarity threshold $\tau$ on inference accuracy, retrieval frequency, and latency. Subplots (a-f) show the accuracy on various long-input and long-output benchmarks. Subplots (g-h) illustrate the corresponding increase in retrieval frequency, and subplot (i) shows the resulting impact on inference latency. The results demonstrate the trade-off between accuracy and efficiency governed by $\tau$.

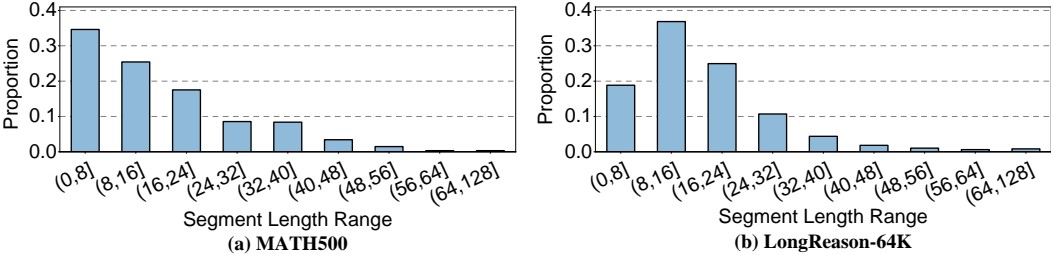

Figure 15: **Distribution of semantic segment lengths.** Histograms for (a) MATH500 and (b) LongReason-64K. The long-tailed distribution indicates that reasoning tasks frequently form long stable semantic segments, which allows LouisKV to effectively amortize the retrieval overhead by skipping redundant operations during these stable periods.

h), the number of retrievals increases with $\tau$ for both task types. Since each retrieval introduces non-negligible overhead, the inference latency consequently increases as well. We measured the average per-token inference latency for different $\tau$ values in a long-input, long-output (32K+16K) scenario using Qwen3-8B, as depicted in Figure 14(i). The results show that for Qwen3-8B, $\tau = 0.7$ represents an ideal balance, achieving significant efficiency gains while maximally preserving inference accuracy.

### D.5 DISTRIBUTION OF SEMANTIC SEGMENT LENGHTS.

To substantiate our amortization claim with hard numbers, we perform a detailed analysis using the Qwen3-8B model with the default threshold $\tau = 0.7$.

We first measure the average semantic segment length across different tasks. Reasoning tasks like MATH500 and GPQA exhibit an average segment length of approximately 14-18 tokens. This implies that LouisKV reduces the retrieval frequency by more than $10\times$ compared to per-token methods. Even for long-document QA tasks (e.g., NarrativeQA) which involve more frequent topic shifts, the average length remains around 3-6 tokens. This provides strong evidence that LouisKV significantly amortizes the retrieval cost across diverse scenarios.

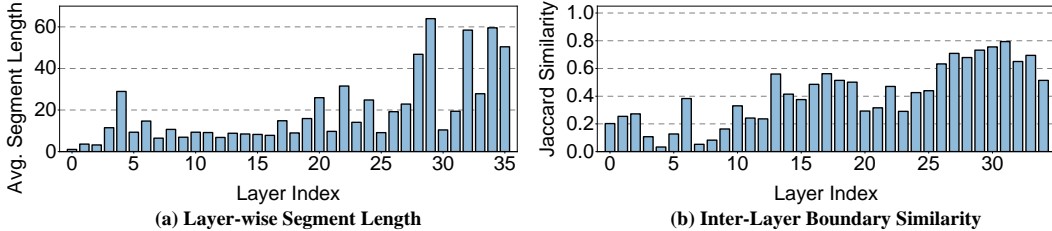

Figure 16: **Per-layer analysis of semantic segmentation.** (a) Average semantic segment length at each Transformer layer. Shallow layers (e.g., 0-10) exhibit shorter segments while deep layers (e.g., 28-34) form longer segments. (b) Jaccard similarity of boundary trigger indices between adjacent layers. The low similarity across most layers indicates that semantic boundaries are detected at different points, validating the necessity of our per-layer retrieval strategy.

We further visualize the full distribution of segment lengths in Figure 15 of the paper. The histograms illustrate a long-tailed distribution for reasoning tasks. While short segments are present, a significant proportion of the generation steps fall into long segments (spanning 16 to 128 tokens). These long segments play a crucial role in reducing retrieval overhead, thereby contributing substantially to the overall end-to-end speed improvement.

### D.6 PER-LAYER BOUNDARY ANALYSIS

To provide a rigorous justification for our layer-wise retrieval design, we analyze the similarity of semantic boundary detection across different Transformer layers. The analysis is conducted using the Qwen3-8B model on the MATH500 dataset.

**Layer-wise Segment Length.** We first measure the average length of semantic segments at each layer. As shown in Figure 16(a) of the paper, shallow layers (Layers 0-3) exhibit shorter average segments (approximately 4-6 tokens). This observation aligns with the intuition that lower layers capture lower-level, rapidly changing features such as syntactic structures or lexical choices, thus identifying more frequent semantic boundaries. In stark contrast, deep layers (e.g., Layers 28-34) exhibit substantially longer segments. This reflects their function in processing high-level abstract concepts, which remain stable during extended reasoning chains.

**Inter-Layer Similarity on Boundaries.** To further quantify the similarity in trigger points across layers, we calculate the Jaccard similarity of the boundary-triggering token indices between adjacent layers. As illustrated in Figure 16(b), the similarity is notably low, especially in the shallow layers. This low similarity indicates that a semantic shift occurring in a lower layer does not necessarily result in a corresponding shift in a higher layer. This phenomenon validates the necessity of our per-layer segmentation strategy, which enables each layer to independently manage the retrieval and segmentation frequencies.

## E USE OF LLMS

In this research, we utilized Large Language Models (LLMs) as an auxiliary tool to enhance research efficiency and the quality of the manuscript. The authors maintained strict oversight throughout this process, ensuring adherence to the principles of academic integrity. We affirm that the authors bear full and final responsibility for all content, analyses, and conclusions presented herein. A detailed breakdown of our specific use cases for these models is provided below:

### LANGUAGE POLISHING AND COPY-EDITING

We utilized LLMs to assist with language refinement in this manuscript. Its use was confined to improving grammar, clarity, and overall readability. The authors reviewed all suggested changes to ensure the integrity of our original arguments and that the scientific content remained unaltered.

LITERATURE REVIEW AND BRAINSTORMING

In the preliminary stages of research, an LLM was used to summarize established concepts and survey related work, which helped in identifying potential research gaps. However, all final literature citations and core research ideas were independently developed, verified, and validated by the authors.

