# OpenReview forum: "LouisKV: Efficient KV Cache Retrieval for Long Input-Output Sequences"
_ICLR.cc/2026/Conference — ICLR 2026 Poster_

### Official Review · Reviewer_7kuj · 2025-10-31

**Soundness:** 3
**Presentation:** 3
**Contribution:** 2
**Rating:** 4
**Confidence:** 4

**Summary:**

The paper introduces LouisKV, a KV cache management framework for long-context and long-generation inference. The design is motivated from the observation of temporal locality between adjacent tokens and distinct spatial distributions between the long inputs and long outputs. During prefill, LouisKV uses k-means clusters and triggers retrieval during decoding only at semantic boundaries detected by query vector cosine similarity. Offloading is also performed for efficiency, and additional system level optimizations using CUDA kernels is done to enable speed-ups. Experiments on long-input tasks like long bench and complex reasoning tasks like AIME show that LouisKV achieves a speedup and performs better than other retrieval methods like Arkvale and Quest.

**Strengths:**

1. The decoupled management scheme which applies different strategies for both prefill and decoding stage, along with system level optimizations lead to strong performance compared to the other testes retrieval baselines.
2. Experimental setup is very comprehensive, testing for not only long-context input, which is more common in KV cache literature, but also long-output using reasoning benchmarks like MATH500 and AIME.

**Weaknesses:**

The contribution of the paper is spread across multiple aspects, making it difficult to recognize the main contribution. LouisKV modifies multiple stages of the LLM inference pipeline: Prefill with k-means clustering and asynchronous offloading. Decoding with semantic-boundary detection, and asynchronous offloading. System level optimizations with triton kernels. While each piece makes sense, it is difficult to assess conceptual novelty.

For instance, the two observations noted in motivation has been reported before and already made use of in prior works. For example, ShadowKV [1] makes use of temporal locality to rebuild only necessary KV pairs. The second observation of distinct patterns has also been noted in SCOPE [2]. Similarly, the novelty of boundary trigger is limited, as FreeKV [3] measures cosine similarity between adjacent tokens' query vectors during long generation, and clustering has also been used during prefill stage (ClusterKV [4]). While the synthesis of these components is novel and the engineering effort is appreciated, it is difficult to isolate or have a view of what the main contribution of the paper is.


[1] Sun, Hanshi, et al. "Shadowkv: Kv cache in shadows for high-throughput long-context llm inference." arXiv preprint arXiv:2410.21465 (2024).

[2] Wu, Jialong, et al. "Scope: Optimizing key-value cache compression in long-context generation." arXiv preprint arXiv:2412.13649 (2024).

[3] Liu, Guangda, et al. "FreeKV: Boosting KV Cache Retrieval for Efficient LLM Inference." arXiv preprint arXiv:2505.13109 (2025).

[4] Liu, Guangda, et al. "Clusterkv: Manipulating llm kv cache in semantic space for recallable compression." 2025 62nd ACM/IEEE Design Automation Conference (DAC). IEEE, 2025.

**Questions:**

1. How sensitive are results to the local buffer W and sink tokens S?

2. How was the number of clusters determined in k-means?

3. The clustering on prefill KVs can be quite expensive. Could you quantify this latency overhead and how it scales with input length?

---

> ### Author Response · Authors · 2025-11-22
>
> Thank you for your valuable comments and detailed analysis of the related work. We will emphasize our core contribution and conceptual novelty in the next revision.
>
> **W1.  The clarification of main contribution and conceptual novelty.**
>
> We understand your concern that LouisKV integrates several techniques (e.g., clustering, similarity-based triggers), which may make the main contribution appear diffuse. **Nevertheless, we respectfully argue that the core contribution of LouisKV lies in the novel KV retrieval architecture that, for the first time, provides a unified and efficient solution for diverse long-sequence inference scenarios, rather than in the individual components.** The design is not a simple synthesis of existing components, but is a holistic solution driven by the novel observations of critical KV access patterns.
>
> A deeper comparison has further strengthened our confidence in LouisKV's unique architecture. Then, we would like to address your concerns regarding conceptual novelty as follows:
>
> 1. **Clarification of Conceptual Novelty.**
>
>    We analyze many prior works and, based on these studies and in-depth analysis, derive a deeper and novel observation: critical KVs exhibit distinct distribution patterns across input and output sequences.
>    Based on this novel observation, **we are the first to propose a decoupled management scheme specifically for KV Retrieval:** using Semantic Clusters for the sparse patterns of the input, and Temporal Segments for the dense patterns of the output. This **"Input-Cluster / Output-Segment" design allows retrieval units to better match the model’s attention patterns,** which is the key factor to maintain high accuracy. Concurrently, **our semantic-aware adaptive retrieval strategy, which aligns with the KV cache partitioning of the output,** is the key to high efficiency.
>
> 2. **Differentiation from SCOPE.**
>
>    - Thank you for pointing out SCOPE. While SCOPE also observes differences between the prefill and decoding phases, **its problem domain, core observation, and resulting design are fundamentally different from those of LouisKV.**
>    - **Different Problem Domains: SCOPE is a KV Dropping method,** where the core challenge is to accurately decide which tokens to permanently discard. **In contrast, LouisKV is a KV Retrieval method,** which preserves the full KV cache and faces the challenge of efficiently and accurately retrieving the most critical KVs during decoding.
>    - **Different Observations and Designs: SCOPE observes that the KV cache of the input is often over-compressed during decoding.** Its design, therefore, is to isolate and preserve the input KV cache. **In contrast, LouisKV's key observation is that critical KVs exhibit distinct distribution patterns across the input and output.** This novel observation led us to a unique "Input-Cluster / Output-Segment" management scheme.
>
> 3. **Novelty Beyond ClusterKV and FreeKV/ShadowKV.**
>
>    - **Beyond ClusterKV:** While ClusterKV explores clustering only for the prefill cache, LouisKV's novelty stems from observing the distinct distribution patterns of input and output KVs. **This lead to our creative, decoupled "Input-Cluster / Output-Segment" scheme, which retains clustering for the input's sparse patterns while introducing a novel segmentation strategy for the output's dense patterns.**
>    - **Beyond FreeKV/ShadowKV:** While prior works explore the concepts of “temporal locality” and “query vector’s similarity”, the novelty of LouisKV lies in the unique application of these concepts. LouisKV not only uses this for efficient triggering but, more critically, **creatively uses the drop in query similarity to dynamically define the boundaries of the temporal segments** for the output KV cache.
>
> A unified KV retrieval framework applicable to diverse long-sequence scenarios is urgently needed by the LLM community. Our work aims to provide such a well-validated solution. To achieve this, we conduct extensive validation studies and optimize every possible aspect of the design. Instead of highlighting the contribution of a single component, our paper focuses on establishing a solid and holistic system, which serves as its core contribution. We will expand our discussion in the final version to include these additional studies, better positioning LouisKV’s unique contribution within this evolving research landscape.  We believe they complement rather than diminish the novelty of our approach.

---

> ### Author Response · Authors · 2025-11-22
>
> **Q1. How sensitive are results to the local buffer W and sink tokens S?**
>
> The selection of S and W has a relatively low impact on inference efficiency but significantly affects inference accuracy. Therefore, We conduct a detailed sensitivity analysis for the local buffer size W and the number of sink tokens S.
>
> 1. Long-Input Understanding Tasks.
>
>    We set the local buffer W=128 and vary the number of sink tokens S, evaluating performance on three representative datasets from LongBench.  As shown in the first table below, increasing S from 0 to 32 yields a better score. This suggests that preserving a small amount of initial tokens is sufficient for a promising score. We then set S=32 and vary the size of W. The results, shown in the second table below, indicate that a moderately-sized buffer is sufficient to capture the most critical recent context. This highlights the importance of retaining recent context.
>
>    | Sink Token (S) | NarrativeQA | Qasper | MultifieldQA |
>    | -------------- | ----------- | ------ | ------------ |
>    | 0              | 22.54       | 40.1   | 48.52        |
>    | 16             | 26.45       | 42.25  | 53.19        |
>    | 32             | 27.53       | 44.31  | 54.22        |
>    | 64             | 27.33       | 43.38  | 54.41        |
>    | 128            | 26.62       | 43.8   | 55.78        |
>
>    | Buffer Size (W) | NarrativeQA | Qasper | MultifieldQA |
>    | --------------- | ----------- | ------ | ------------ |
>    | 0               | 23.76       | 41.1   | 47.24        |
>    | 16              | 24.76       | 40.15  | 51.76        |
>    | 32              | 28.29       | 43.25  | 55.04        |
>    | 64              | 28.51       | 43.53  | 54.51        |
>    | 128             | 27.53       | 44.31  | 54.22        |
>
> 2. Long-Output Reasoning Tasks.
>
>    We first set the local buffer W to 128 and vary the number of sink tokens S. Then, we fix S at 64 and adjust the size of W. We sample 100 questions from MATH500 and LongReason32K, respectively.
>
>    As shown in the tables below, larger values for both S and W tend to yield better accuracy on long-reasoning tasks. For instance, MATH500 performs best with S=512, and LongReason-32K peaks at W=256. This is intuitive, as complex reasoning often requires both anchoring to the original problem definition (requiring a large S) and referencing a long chain of intermediate steps (requiring a large W).
>
>    | Sink Token (S) | MATH500 | LongReason-32K |
>    | -------------- | ------- | -------------- |
>    | 32             | 0.87    | 0.68           |
>    | 64             | 0.86    | 0.67           |
>    | 128            | 0.87    | 0.71           |
>    | 256            | 0.88    | 0.7            |
>    | 512            | 0.89    | 0.73           |
>
>    | Buffer Size (W) | MATH500 | LongReason-32K |
>    | --------------- | ------- | -------------- |
>    | 32              | 0.83    | 0.62           |
>    | 64              | 0.84    | 0.64           |
>    | 128             | 0.86    | 0.67           |
>    | 256             | 0.89    | 0.73           |
>    | 512             | 0.91    | 0.72           |
>
> **In summary, our analysis reveals that for long-input understanding tasks, a moderate S (e.g., 32) and a moderate W (e.g., 64) are an effective combination; for long-output reasoning tasks, larger S and W are generally more beneficial.**

---

> ### Author Response · Authors · 2025-11-22
>
> **Q2. How was the number of cluster determined in K-means?**
>
> Our method does not directly set the number of clusters. Instead, we define the **average cluster size** and dynamically calculate the number of clusters using the following formula:
>
> > **Number of Clusters = Input Sequence Length / Average Cluster Size**
>
> This approach allows us to flexibly adjust the number of clusters based on the input sequence length. To determine the optimal average cluster size, we conducted experiments from two perspectives:
>
> 1. Impact of Average Cluster Size on Inference Accuracy
>
>    We evaluate the impact of different average cluster sizes (16, 32, 64, 128) on inference accuracy across several long-input tasks. The results are summarized below:
>
>    | Average Cluster Size | NarrativeQA | Qasper | MultifieldQA | LongReason-32K |
>    | -------------------- | ----------- | ------ | ------------ | -------------- |
>    | 16                   | 27.53       | 44.31  | 54.22        | 0.73           |
>    | 32                   | 27.14       | 42.77  | 53.88        | 0.71           |
>    | 64                   | 26.82       | 41.58  | 52.57        | 0.64           |
>    | 128                  | 24.25       | 39.66  | 50.21        | 0.66           |
>    | FullCache Acc.       | 28.26       | 44.38  | 55.27        | 0.74           |
>
>    As shown in the table, increasing the average cluster size led to a decline in inference accuracy across all datasets.
>
> 2. Impact of Average Cluster Size on Inference Efficiency
>
>    We further evaluate the impact of different average cluster sizes on end-to-end inference latency across two long-sequence scenarios: 32K+512 (long-input, short-output) and 32K+16K (long-input, long-output). The results are summarized below:
>
>    | Average Cluster Size | 32K+512 (s) | 32K+16K (s) |
>    | -------------------- | ----------- | ----------- |
>    | 16                   | 39.12       | 732         |
>    | 32                   | 38.24       | 726         |
>    | 64                   | 38.71       | 728         |
>    | 128                  | 37.85       | 722         |
>
>    The results indicate that increasing the average cluster size slightly reduces end-to-end inference latency. This is expected, as managing and retrieving fewer, larger clusters reduces retrieval overhead.
>
> While larger clusters (e.g., 128) can achieve approximately a 1% reduction in inference latency, they result in **significant** accuracy degradation (e.g., a 5% drop on Qasper). **To balance inference efficiency and accuracy, we select 16 as the default average cluster size. This setting also aligns with existing KV retrieval methods (e.g., Quest and Arkvale), which set their block size to 16, ensuring a fair comparison.**

---

> ### Author Response · Authors · 2025-11-22
>
> **Q3. The Latency Overhead of Clustering.**
>
> We quantify the clustering overhead from two perspectives: 1) its impact on TTFT across different input lengths; and 2) the standalone overhead of the clustering operation itself.
>
> 1. TTFT Across Different Input Lengths
>
>    We first evaluate the TTFT of LouisKV compared to FullCache under various input lengths. The results are shown below:
>
>    | Input Length       | 1K    | 2K    | 4K    | 8K    | 16K  | 32K   | 64K   |
>    | ------------------ | ----- | ----- | ----- | ----- | ---- | ----- | ----- |
>    | LouisKV TTFT (s)   | 0.187 | 0.366 | 0.758 | 1.597 | 3.513 | 8.85 | 23.49 |
>    | FullCache TTFT (s) | 0.178 | 0.359 | 0.724 | 1.521 | 3.398 | 8.61 | 22.81 |
>
>    As shown in the table, the TTFT of LouisKV is only marginally higher than that of FullCache, even for shorter inputs (e.g., 1K, 2K). **This minor increase is due to our Triton-optimized K-means clustering algorithm and the asynchronous execution of clustering during the prefill stage, which effectively hides the clustering overhead within the forward computation.**
>
> 2. Standalone Clustering Latency Overhead.
>
>    To further quantify the clustering overhead, we measure the latency overhead introduced by the clustering operation in isolation. We first set the number of clustering iterations to 10 and measure the clustering latency for different input lengths. The results are shown below:
>
>    | Input Length | 1K    | 2K    | 4K    | 8K    | 16K   | 32K   | 64K  |
>    | ------------ | ----- | ----- | ----- | ----- | ----- | ----- | ---- |
>    | Latency (s)  | 0.055 | 0.060 | 0.108 | 0.159 | 0.272 | 0.776 | 2.34 |
>
>    We then set the input length to 16K and vary the number of clustering iterations. The results are shown below:
>
>    | Iteration   | 1     | 5     | 10    | 20    | 50   | 100  |
>    | ----------- | ----- | ----- | ----- | ----- | ---- | ---- |
>    | Latency (s) | 0.069 | 0.155 | 0.272 | 0.473 | 1.13 | 2.23 |
>
>    **The results demonstrate that the clustering overhead scales with both input length and the number of iterations. In our paper, we configure the clustering process with 10 iterations. Crucially, due to our asynchronous pipeline design, this overhead is effectively hidden during the prefill computation, resulting in negligible impact on the overall TTFT.**

---

### Official Review · Reviewer_cVfv · 2025-10-31

**Soundness:** 3
**Presentation:** 3
**Contribution:** 3
**Rating:** 6
**Confidence:** 4

**Summary:**

The paper proposes **LouisKV**, a KV-cache retrieval framework for long input and long output generation. The core idea is to avoid per-token retrieval by exploiting temporal locality in decoding and to improve precision by managing input and output KV caches differently. Concretely, LouisKV (i) detects semantic boundaries using the cosine similarity of consecutive query vectors and only triggers retrieval at these boundaries, and (ii) applies k-means clustering over prefill keys for inputs while forming temporal segments for generated outputs, each represented by a centroid for fast similarity lookup. The system adds Triton and CUDA kernels and a group-consistent selection strategy to reduce transfer and compute overhead.

**Strengths:**

1) **Decoding efficiency by fewer retrieval triggers.** Triggering only at semantic boundaries reduces repeated estimation and transfers while preserving accuracy within a segment, which the paper motivates and supports with locality measurements and latency breakdown.

2) **Covers multiple inference scenarios.** The study includes long-input short-output, short-input long-output, and long-input long-output, showing stable accuracy at fixed budgets and scale-up in batch size without OOM.

3) **Solid systems work.** Custom Triton and CUDA kernels and group-consistent selection improve throughput in realistic deployments with GQA. The ablation indicates which component contributes most.

**Weaknesses:**

1) **Motivation scope.** The assumption that decoding naturally forms stable semantic segments holds on math and step-wise tasks but may not on dialog or code completion with frequent topic shifts.

2) **Clustering cost and TTFT.** Input clustering is central, yet the paper reports end-to-end latency without an explicit time-to-first-token under very long prompts. Please add TTFT curves vs prompt length and show the amortization benefit when queries are short. Also report k-means wall time vs input length and number of clusters, and whether the asynchronous pipeline hides this cost under prefill compute.

3) **Long-context stress tests.** Results include LongReason up to 64K. RULER is a standard stress suite for long-context retrieval and sequence length extrapolation. Please add RULER at 32K-128K with accuracy and latency, and clarify memory usage under those settings.

**Questions:**

1) **Trigger statistics.** Please report the distribution of semantic segment lengths across datasets, average r at boundaries, and per-layer agreement on boundary detection. This can justify the amortization claim with hard numbers beyond the ablation.

2) **Generated KV retention in long-reasoning.** In short-input long-output tasks, do you offload all generated KVs to CPU after the local buffer fills so they remain recallable. If yes, what is the incremental CPU-GPU traffic per segment, and how does that compare with page-level Arkvale under the same budget. A per-token bytes-moved metric would help.

3) **Cluster locality in text space.** Within a cluster, are source text spans mostly contiguous or scattered. If clusters often map to contiguous spans, would span-aware grouping or block-sparse layouts reduce transfers further. Any analysis of edit distance between token positions in the same cluster.

4) **Citation coverage.** More semantic-based retrieval work could be added.
[1] Chen, Guoxuan, et al. "Sepllm: Accelerate large language models by compressing one segment into one separator." arXiv preprint arXiv:2412.12094 (2024).
[2] Zhu, Yuxuan, et al. "SentenceKV: Efficient LLM Inference via Sentence-Level Semantic KV Caching." arXiv preprint arXiv:2504.00970 (2025).
[3] Hooper, Coleman, et al. "Squeezed attention: Accelerating long context length llm inference." arXiv preprint arXiv:2411.09688 (2024).

---

> ### Author Response · Authors · 2025-11-28
>
> Thanks for your insightful review and valuable feedback. Your recognition of our strengths is a great encouragement, and the weaknesses and questions you raised are crucial for us to further improve our work.  We have conducted in-depth analysis and supplementary experiments based on your comments and would like to respond to each point below.
>
> **Note on Revision:** We have uploaded a revised version of the paper. **Major updates regarding Trigger Statistics (Q1) and Citation Coverage (Q4) are highlighted in blue on Pages 4, 18, and 19.** We strongly recommend reviewing these highlighted sections along with the response below, as they provide better clarification for your concerns regarding Q1 and Q4.
>
> **W1. Motivation Scope.**
>
> We respectfully clarify that our motivation stems from the **universality of temporal locality** in auto-regressive decoding, rather than an assumption that all tasks must exhibit long reasoning chains. Even in tasks with frequent topic shifts (e.g., dialogue), decoding proceeds in semantic units (e.g., phrases or short sentences) which **constitute stable, shorter semantic segments.** Thus, the assumption of "stable semantic segments" holds across tasks, varying only in granularity.
>
> 1. **Adaptive Robustness**
>
>    LouisKV detects semantic boundaries by monitoring the cosine similarity of query vectors. In scenarios with frequent topic shifts (e.g., dialogue), the query similarity drops below the threshold $\tau$ more repeatedly, correctly reflecting the finer semantic granularity.  Our experiments encompass a diverse range of tasks, including strict step-by-step reasoning (e.g., MATH500) and long-document QA (e.g., NarrativeQA)**.** The latter serves as an effective proxy for dialogue-like scenarios, as it similarly involves shorter generated content and frequent context switching without rigid reasoning chains.
>
>    As detailed in our response to **Q1**, our empirical statistics show average segment lengths of **14.5-18.2 tokens for reasoning tasks** and **3.6-6.3 tokens for non-step-wise long-document QA tasks.** This proves that our motivation (existence of semantic segments) holds even in dynamic settings, and LouisKV accurately identifies these intrinsic differences to ensure the KV cache is retrieved in time to preserve accuracy.
>
> 2. **Efficiency in Dynamic Scenarios**
>
>    Even in tasks with frequent shifts, LouisKV retains a significant efficiency advantage. For instance, retrieving once every 4 tokens (in a dialogue scenario) still reduces the retrieval overhead by 75% compared to expensive per-token approaches. This is empirically validated in **Figure 6(a) of the paper**, where LouisKV achieves up to **1.8x speedup** in short ouput tasks and **4.7x speedup** in long output tasks.
>
>    Consequently, for tasks involving frequent topic shifts, the theoretical speedup would logically fall within this significant range (likely trending towards the 1.8x lower bound, which is still a substantial improvement). This evidence demonstrates that our method effectively amortizes retrieval costs even when semantic segments are relatively short. **Therefore, our motivation remains valid and LouisKV delivers clear performance benefits across dynamic scenarios.**

---

> ### Author Response · Authors · 2025-11-28
>
> **W2. Clustering Cost and TTFT.**
>
> We address the concern regarding clustering overhead by quantifying: 1) its impact on TTFT across different input lengths; and 2) the standalone overhead of the clustering operation itself.
>
> 1. TTFT Across Different Input Lengths
>
>    We compare the TTFT of LouisKV against FullCache across various input lengths. As shown in the table below, the TTFT of LouisKV is only marginally higher than that of FullCache, even for shorter inputs (e.g., 1K, 2K). **This negligible increase is due to our Triton-optimized K-means clustering algorithm and the asynchronous execution of clustering during the prefill stage, which effectively hides the clustering overhead within the forward computation.**
>
>    | Input Length       | 1K    | 2K    | 4K    | 8K    | 16K   | 32K  | 64K   |
>    | ------------------ | ----- | ----- | ----- | ----- | ----- | ---- | ----- |
>    | LouisKV TTFT (s)   | 0.187 | 0.366 | 0.758 | 1.597 | 3.513 | 8.85 | 23.49 |
>    | FullCache TTFT (s) | 0.178 | 0.359 | 0.724 | 1.521 | 3.398 | 8.61 | 22.81 |
>
> 2. Standalone Clustering Latency Overhead
>
>    To isolate the cost of clustering overhead, we measure its standalone latency. We first set the number of clustering iterations to 10 and measure the clustering latency for different input lengths. The results are shown below:
>
>    | Input Length | 1K    | 2K    | 4K    | 8K    | 16K   | 32K   | 64K  |
>    | ------------ | ----- | ----- | ----- | ----- | ----- | ----- | ---- |
>    | Latency (s)  | 0.055 | 0.060 | 0.108 | 0.159 | 0.272 | 0.776 | 2.34 |
>
>    We also analyze sensitivity to the number of iterations. We set the input length to 16K and vary the number of clustering iterations. **In our paper, we configure the clustering process with 10 iterations.**
>
>    | Iteration   | 1     | 5     | 10    | 20    | 50   | 100  |
>    | ----------- | ----- | ----- | ----- | ----- | ---- | ---- |
>    | Latency (s) | 0.069 | 0.155 | 0.272 | 0.473 | 1.13 | 2.23 |
>
>    **The results demonstrate that the clustering overhead scales with both input length and the number of iterations. Crucially, due to our asynchronous pipeline design, this overhead is effectively hidden during the prefill computation, resulting in a negligible impact on the overall TTFT, which increases by an average of just 3.7% and never more than 5.1% in our tests.**

---

> ### Author Response · Authors · 2025-11-28
>
> **W3. Long-Context Stress Tests.**
>
> We conduct comprehensive stress tests on the RULER dataset across context lengths of 32K, 64K, and 128K to evaluate accuracy, latency, and memory usage. We compare LouisKV against **FullCache** and the state-of-the-art KV retrieval method **Arkvale**. For both LouisKV and Arkvale, the KV cache budget is consistently set to 2K tokens across all datasets and sequence lengths for the accuracy evaluation. All tests are run on a single A6000 GPU with 48 GB of GDDR6 DRAM, as specified in our main paper.
>
> 1. **Accuracy Analysis**
>
>     We evaluate performance across 13 datasets from the RULER suite. As shown in the tables below, LouisKV demonstrates strong performance in extreme long-context scenarios. At 32K and 64K, its accuracy is **comparable to the lossless FullCache baseline**. Crucially, at **128K, FullCache fails with an Out-of-Memory (OOM) error** on our hardware, highlighting the fundamental necessity of retrieval-based methods in such extreme contexts. In contrast, LouisKV completes the task successfully. Furthermore, LouisKV **significantly and consistently outperforms Arkvale** across all tested lengths (32K, 64K, and 128K). This performance gap underscores the effectiveness of our design in accurately identifying and retrieving the most critical KV entries.
>
>     - **Accuracy on RULER at 32K**
>
>         | Method | niah_single_1 | niah_single_2 | niah_single_3 | niah_multikey_1 | niah_multikey_2 | niah_multikey_3 | niah_multivalue | niah_multiquery | vt | cwe | fwe | qa_1 | qa_2 | Avg. |
>         | --- | --- | --- | --- | --- | --- | --- | --- | --- | --- | --- | --- | --- | --- | --- |
>         | LouisKV | 100 | 100 | 100 | 100 | 100 | 99.5 | 96.12 | 99.75 | 99.5 | 49.85 | 92.67 | 75 | 55.5 | 89.84 |
>         | Arkvale | 100 | 99.5 | 99.5 | 99.5 | 99.5 | 92 | 94.12 | 99.75 | 99.2 | 31.3 | 92.17 | 73.5 | 54 | 87.23 |
>         | FullCache | 100 | 100 | 100 | 100 | 100 | 100 | 97.12 | 99.75 | 99.8 | 52 | 95.33 | 74.5 | 54 | 90.19 |
>     - **Accuracy on RULER at 64K**
>
>         | Method | niah_single_1 | niah_single_2 | niah_single_3 | niah_multikey_1 | niah_multikey_2 | niah_multikey_3 | niah_multivalue | niah_multiquery | vt | cwe | fwe | qa_1 | qa_2 | Avg. |
>         | --- | --- | --- | --- | --- | --- | --- | --- | --- | --- | --- | --- | --- | --- | --- |
>         | LouisKV | 100 | 100 | 100 | 100 | 98.5 | 92 | 96.88 | 100 | 96.5 | 11.8 | 85.33 | 72 | 50 | 84.85 |
>         | Arkvale | 100 | 98.5 | 99 | 99.5 | 96.5 | 57.5 | 94.5 | 99.75 | 95.2 | 2.35 | 78.17 | 71.5 | 49 | 80.11 |
>         | FullCache | 100 | 100 | 100 | 100 | 99 | 95.5 | 97.62 | 99.75 | 98.6 | 12.05 | 86.17 | 72 | 49 | 85.36 |
>     - **Accuracy on RULER at 128K**
>
>         | 128K | niah_single_1 | niah_single_2 | niah_single_3 | niah_multikey_1 | niah_multikey_2 | niah_multikey_3 | niah_multivalue | niah_multiquery | vt | cwe | fwe | qa_1 | qa_2 | Avg. |
>         | --- | --- | --- | --- | --- | --- | --- | --- | --- | --- | --- | --- | --- | --- | --- |
>         | LouisKV | 100 | 100 | 100 | 98 | 79 | 25 | 90.5 | 96.62 | 80.7 | 0.1 | 62.33 | 69 | 44 | 72.71 |
>         | Arkvale | 100 | 99.5 | 97.5 | 96 | 73 | 6.5 | 72 | 90.62 | 79.5 | 0 | 60.17 | 68 | 45 | 68.29 |
>         | FullCache | OOM | OOM | OOM | OOM | OOM | OOM | OOM | OOM | OOM | OOM | OOM | OOM | OOM | OOM |
> 2. **Latency and Memory Analysis**
>
>     To evaluate system performance, we measure end-to-end latency and peak memory usage across different input lengths while generating 100 output tokens. As shown in the table below, LouisKV **consistently outperforms** **Arkvale** in terms of latency, owing to its more efficient clustering and retrieval mechanism. Notably, LouisKV ****achieves latency **slightly lower than FullCache.** This demonstrates that the reduction in attention computation cost  effectively offsets the overhead of retrieval. This efficiency advantage becomes even **more pronounced during longer generation tasks**, as detailed in Section 5.3 of our main paper.
>
>     | E2E Latency (s) | 32K | 64K | 128K |
>     | --- | --- | --- | --- |
>     | LouisKV | 12.72 | 29.69 | 78.59 |
>     | Arkvale | 18.33 | 38.64 | 85.62 |
>     | FullCache | 13.59 | 30.59 | OOM |
>
>     The memory analysis reveals a critical advantage of retrieval-based methods for long-context inference. As shown in the table below, **FullCache results in an OOM error at the 128K sequence length** on our 48GB GPU. In contrast, both LouisKV and Arkvale operate comfortably by retaining only a small, critical portion of the KV cache on the GPU and offloading the rest to CPU memory. This design makes them highly scalable solutions for scenarios of exrtemely long context lengths and large batch sizes where FullCache fundamentally fails.
>     | Memory Usage (GB) | 32K | 64K | 128K |
>     | ----------------- | ----- | ----- | ----- |
>     | LouisKV  | 22.85 | 24.51 | 32.45 |
>     | Arkvale| 23.82 | 29.17 | 35.84 |
>     | FullCache| 24.91 | 34.35 | OOM   |

---

> ### Author Response · Authors · 2025-11-28
>
> **Q1. Trigger Statistics.**
>
> To substantiate our amortization claim with hard numbers, we perform a detailed analysis using the Qwen3-8B model with the default threshold $\tau=0.7$. **We have incorporated these detailed statistics, along with the corresponding visualizations (Figure 14 and Figure 15), into Appendix D.5 and D.6 of the revised paper.**
>
> 1. **Distribution of Semantic Segment Lengths.**
>
>    We first measure the average length of semantic segments across different tasks. As shown in the table below, for reasoning tasks like MATH500 and GPQA, the average segment length is approximately **14-18 tokens**. This implies that LouisKV reduces the retrieval frequency by **more than 10 $\times$** compared to per-token methods. Even for long-document QA tasks (e.g., NarrativeQA), which involve more frequent topic shifts, the average length remains around **3-6 tokens**. This provides strong evidence that LouisKV significantly amortizes the retrieval cost across diverse scenarios.
>
>    |                              | NarrativeQA | Qasper | MultifieldQA | MATH500 | GPQA | AIME  | LongReason-64K |
>    | ---------------------------- | ----------- | ------ | ------------ | ------- | ---- | ----- | -------------- |
>    | Avg. Segment Length (tokens) | 4.84        | 3.57   | 6.26         | 14.51   | 17.1 | 16.67 | 18.24          |
>
>    We further visualize the full distribution of segment lengths in **Figure 14** of the paper. The histograms illustrate a **long-tailed distribution** for reasoning tasks. While short segments are present, a significant proportion of the generation steps fall into long segments (spanning 16 to 128 tokens). These long segments play a crucial role in reducing retrieval overhead, thereby contributing substantially to the overall end-to-end speed up.
>
> 2. **Boundary Similarity Score & $\tau$ Sensitivity.**
>
>    We further analyze how the threshold **$\tau$** impacts the segment length and the average similarity r at the detected boundaries. As illustrated in the table below (using MATH500), a higher **$\tau$** imposes a stricter coherence constraint, leading to shorter segments. The "Avg. r at Boundary" column confirms that retrieval is triggered precisely when the semantic similarity drops just below the defined threshold.
>
>    | Threshold (**$\tau$**) | 0.6  | 0.7 (Default) | 0.8  | 0.9  |
>    | ---------------------- | ---- | ------------- | ---- | ---- |
>    | Avg. Segment Length    | 33.3 | 14.51         | 4.34 | 1.31 |
>    | Avg. r at Boundary     | 0.53 | 0.64          | 0.76 | 0.86 |
>
> 3. **Per-Layer Analysis of Semantic Segmentation.**
>
>    To provide a rigorous justification for our layer-wise retrieval design, we analyze the similarity of semantic boundary detection across different Transformer layers. The analysis is conducted using the Qwen3-8B model on the MATH500 dataset.
>
>    - **Layer-wise Segment Length:** We first measure the average length of semantic segments at each layer. As shown in **Figure 15(a)** of the paper, shallow layers (Layers 0-3) exhibit shorter average segments (approximately 4-6 tokens). This observation aligns with the intuition that lower layers capture lower-level, rapidly changing features such as syntactic structures or lexical choices, thus identifying more frequent semantic boundaries. In stark contrast, **deep layers** (e.g., Layers 28-34) exhibit substantially longer segments. This reflects their function in processing high-level abstract concepts, which remain stable during extended reasoning chains.
>    - **Inter-Layer Similarity on Boundaries:** To further quantify the similarity in trigger points across layers, we calculate the Jaccard similarity of the boundary-triggering token indices between adjacent layers. As illustrated in **Figure 15(b)**, the similarity is notably low in the shallow layers. This low similarity indicates that a semantic shift occurring in a lower layer does not necessarily result in a corresponding shift in a higher layer. **This phenomenon validates the necessity of our per-layer segmentation strategy**, which enables each layer to independently manage the retrieval and segmentation frequencies.

---

> ### Author Response · Authors · 2025-11-28
>
> **Q2. Generated KV Retention in Long-Reasoning.**
>
> When the local buffer on the GPU becomes full, LouisKV offloads the KV cache of **only the oldest semantic segment** (not all generated KVs) to the CPU memory pool, ensuring that generated KVs remain recallable. Regarding the incremental CPU-GPU traffic, we use the metric "**bytes-moved-per-token**" (the average amount of KV cache loaded from CPU to GPU per generated token) for comparison. We take the LongReason-64K task, the Qwen3-8B model, and a KV cache budget of 1024 tokens as an example:
>
> - **Arkvale:** As a per-token retrieval method, Arkvale must select and transfer critical KV cache pages at every decoding step. Our measurements indicate that this results in an average KV cache recall of **35.13 MB per generated** **token.**
> - **LouisKV:** Due to our adaptive retrieval strategy, a retrieval operation is triggered only once every 18.2 tokens on average. This drastically reduces the average KV cache transfer to **7.91 MB per token**. This comparison demonstrates that LouisKV reduces the CPU-GPU communication traffic by **over 77%**, effectively alleviating the bandwidth bottleneck in long-output scenarios.
>
> **Q3. Cluster Locality in Text Space.**
>
> To investigate the spatial distribution of tokens within the semantic clusters, we conduct an analysis using the Qwen3-8B model. We sample 50 examples from each of the long-input understanding tasks, with the average cluster size set to 16.
>
> - **Average Continuous Span Count.**
>
>   We introduce the metric "**Average Continuous Span Count"** to quantify the spatial locality. This metric measures the number of disjoint token spans that constitute a single cluster. For a cluster size of 16, a span count of 1 indicates perfect contiguity, while 16 indicates complete scattering. As shown in the table below, the tokens in a cluster are derived from an average of **8 to 10 spans**. This indicates that a cluster aggregates semantically related tokens that are **highly scattered** throughout the prompt, **rather than being contiguous.**
>
>   |                            | Qasper | MultifieldQA | HotpotQA | NarrativeQA | 2WikiMQA | GovReport |
>   | -------------------------- | ------ | ------------ | -------- | ----------- | -------- | --------- |
>   | Avg. Continuous Span Count | 8.25   | 9.22         | 7.91     | 10.09       | 10.18    | 9.29      |
>
> - **Average Token Distance.**
>
>   We also calculate the metric ”**Average Token Distance”** (the mean absolute distance between **all pairs of token positions** within the same cluster). We compare this against a **Random Baseline**, which represents the theoretical expected distance if cluster tokens are randomly drawn from the document. As shown in the table below, the average distance between tokens in a cluster is substantial, often spanning hundreds of tokens. This confirms that the captured semantic relationships are indeed **global and long-range**, extending far beyond the scope of a simple fixed-size block.
>
>   | Task                          | Qasper | MultifieldQA | HotpotQA | NarrativeQA | 2WikiMQA | GovReport |
>   | ----------------------------- | ------ | ------------ | -------- | ----------- | -------- | --------- |
>   | Avg. Token Distance (LouisKV) | 313.51 | 363.09       | 379.25   | 874.91      | 319.85   | 429.18    |
>   | Random Baseline (Theoretical) | 1303   | 1543         | 3087     | 5434        | 1709     | 2895      |
>
> **Q4. Citation Coverage.**
>
> We thank the reviewer for pointing out these relevant works. We have incorporated citations for **SepLLM**, **SentenceKV**, and **Squeezed Attention** into **Section 2 (Related Work)** of our revised manuscript. Discussing these works helps to clarify LouisKV's unique position within the landscape of semantic-based optimization:
>
> - **Differentiation from Linguistic-Based Methods:** Unlike **SepLLM** and **SentenceKV,** which rely on **static linguistic boundaries** (e.g., punctuation) to define semantic units, LouisKV employs a **dynamic retrieval trigger** based on query similarity. This allows our method to adapt to semantic shifts that do not align with explicit sentence structures.
> - **Differentiation from Uniform Clustering:** Unlike **Squeezed Attention**, which applies a uniform clustering strategy, LouisKV introduces a **decoupled management scheme** uniquely tailoring clustering for inputs and temporal segmentation for outputs.
>
> We believe that including these recent studies on semantic-based KV cache optimization provides a more comprehensive context for our research and helps to better position LouisKV's contributions within the field.

---

> ### Author Response · Authors · 2025-11-28
>
> **Conclusion.**
>
> We sincerely appreciate your detailed and constructive feedback, which has greatly enhanced the quality of our paper. Your insightful questions prompted us to conduct and incorporate valuable new analyses on layer-wise segmentation, data traffic, long-context stress tests, and so on. We believe these additions significantly strengthen our arguments and make the paper more comprehensive.
>
> These new findings have been thoroughly documented in our responses and integrated into the revised manuscript. Thank you once again for your time and invaluable guidance. We hope our responses and the revised manuscript have fully addressed your concerns, and we would be happy to discuss any further questions you may have.

---

### Official Review · Reviewer_dBjq · 2025-10-31

**Soundness:** 2
**Presentation:** 2
**Contribution:** 1
**Rating:** 2
**Confidence:** 5

**Summary:**

This paper presents LouisKV, a framework to accelerate long-context LLM inference by optimizing KV cache retrieval.

LouisKV uses a semantic-aware retrieval strategy that triggers KV retrieval only at semantic boundariesr to reduce computation and data transfer.
During the prefill stage, LouisKV performs k-means clustering on the KV cache to group semantically similar tokens, using the centroids for retrieval. In the decode stage, it partitions generated KVs into fixed-size temporal segments, retrieving them as clusters when a semantic boundary is detected.

Experiments demonstrate that LouisKV achieves up to 4.7× speedup over state-of-the-art KV retrieval methods such as Arkvale, while maintaining near-lossless accuracy across diverse long-sequence tasks.

**Strengths:**

The paper presents comprehensive experiments across diverse benchmarks, models, and task types to demonstrate that LouisKV achieves near-lossless accuracy while improving inference efficiency.

**Weaknesses:**

This paper exhibits **a severe lack of novelty** and raises potential concerns about overlap with prior work.

In particular, the core KV cache retrieval mechanism in LouisKV appears **nearly identical** to that of ClusterKV \[1\].
Both of them apply k-means clustering to the KV cache during the prefill stage, select retrieval units based on cluster centroids, and manage KV entries generated during decoding separately.
Furthermore, system-level components such as the asynchronous clustering, offloading pipeline and the CUDA kernel for efficient selection have also been introduced in ClusterKV.
As a result, LouisKV does **not present clear methodological or engineering innovation** beyond what has been previously published.
The absence of explicit discussion or experimental comparison against ClusterKV further weakens the paper’s contribution.

The claimed semantic-aware adaptive retrieval also offers **limited novelty**. The notion of exploiting temporal locality in decoding has been explored in prior works \[2, 3, 4\].
For example, the specific strategy of triggering retrieval based on cosine similarity between consecutive query vectors seems essentially the same as in \[3\].
Consequently, the contribution of LouisKV beyond existing literature is unclear.

\[1\] ClusterKV: Manipulating LLM KV Cache in Semantic Space for Recallable Compression

\[2\] HShare: Fast LLM Decoding by Hierarchical Key-Value Sharing

\[3\] FreeKV: Boosting KV Cache Retrieval for Efficient LLM Inference

\[4\] CaliDrop: KV Cache Compression with Calibration

**Questions:**

What are the unique contributions and technical novelties of LouisKV compared to prior KV retrieval frameworks, particularly ClusterKV.

---

> ### Author Response · Authors · 2025-11-24
>
> Thank you for your valuable feedback on our paper. After carefully considering your comments, we would like to clarify several key points to demonstrate LouisKV's novelty and contribution, particularly in contrast to ClusterKV.
>
> **Q0.  Clarifying a Factual Misunderstanding in Your Summary.**
>
> First, we would like to correct a critical factual misunderstanding in your summary. Your summary mentions: "In the decode stage, it partitions generated KVs into **fixed-size** temporal segments...". This is the exact opposite of our design. One of the core innovations of LouisKV is that it does **not use FIXED-SIZE blocks** to partition the KV cache. As stated in Section 4.2 of our paper: "LouisKV leverages the **semantic boundaries** identified in Section 4.1 to partition the generated KV cache into multiple **temporal segments**." This results in **VARIABLE-LENGTH** KV cache segments tailored for long output sequences, which is a significant departure from works like Arkvale and ClusterKV.
>
> **We are concerned that this factual misunderstanding may have influenced your overall judgment of our work in terms of novelty and contributions. We believe that clarifying this core mechanism of "dynamic segmentation" will aid in assessing the novelty of our work more accurately.**

---

> ### Author Response · Authors · 2025-11-24
>
> **W1 & Q1.  Clarification on Novelty and Comparison with ClusterKV**
>
> **We respectfully argue against the statement "LouisKV appears nearly identical to ClusterKV".**  We agree that both frameworks apply clustering to the input sequence (as we cited in Section 4.2). However, we posit that viewing this component as the core mechanism of LouisKV overlooks the essence of our work.
>
> The core innovation of LouisKV is a decoupled KV management scheme tailored for input and output sequences, driven by our novel observations on KV distribution patterns. This scheme introduces variable-length dynamic segmentation for the output sequence based on semantic boundaries, and is integrated with a matching adaptive retrieval strategy that triggers only at these segment boundaries. These designs are fundamentally distinct from those of ClusterKV.
>
> To elucidate these differences, we break them down as follows:
>
> - **Fundamentally Different Motivations and Goals:** ClusterKV is motivated by the observation that **"semantically similar tokens have similar attention weights,"** with the goal of achieving accurate KV retrieval for long-input scenarios. In contrast, LouisKV’s goal is to provide a unified, high-efficiency framework for diverse long-sequence scenarios. Our design is driven by two observations: **(1) critical KVs exhibit distinct distribution patterns across input and output sequences, and (2) strong temporal locality during decoding.**
> - **Fundamentally Different Management Schemes:** ClusterKV employs a **uniform management** approach for both input and output sequences. In contrast, LouisKV proposes a **decoupled scheme**: it uses k-means clustering for the input sequence and introduces a **DYNAMIC SEGMENTATION** strategy for the output sequence. This design better matches the distinct attention patterns of input and output sequences and is the key factor to maintaining high inference accuracy.
> - **Fundamentally Different Retrieval Strategies:** ClusterKV performs **retrieval at every decoding step.** In contrast, LouisKV employs a semantic-aware adaptive retrieval strategy, triggering **retrieval only when the semantic boundary is detected.** This approach significantly reduces retrieval frequency, thereby minimizing computation and data transfer overhead.
> - **A Note on System Optimizations:** We would like to clarify that the engineering optimizations the reviewer mentioned, such as CUDA kernels and asynchronous execution, are **widely recognized as common practices in high-performance systems** within this domain and are **NOT** unique innovations of ClusterKV. Our optimizations are specifically designed to support our unique KV management and retrieval. For instance, we design custom kernels to efficiently retrieve and aggregate **variable-length input clusters and dynamic output segments** and we build an asynchronous pipeline that matches our offload logic. Therefore, we respectfully consider our improvements in these aspects **not an overlap with ClusterKV, but rather a necessary and novel implementation** tailored to our unique methodology.
>
> Finally, to more intuitively illustrate the differences between ClusterKV and LouisKV, we summarize them in the table below:
>
> | Feature                  | ClusterKV                                                    | LouisKV (Our Work)                                           |
> | ------------------------ | ------------------------------------------------------------ | ------------------------------------------------------------ |
> | **Design Motivation**    | Semantically similar tokens have similar attention weights   | **(1) Temporal locality (2) KV distribution patterns differ between input and output** |
> | **Input KV Management**  | K-Means Clustering                                           | K-Means Clustering                                           |
> | **Output KV Management** | No additional design (following input‘s approach)            | **Dynamic Temporal Segmentation** based on semantic boundaries |
> | **Retrieval Trigger**    | No additional design (retrieval at each token following prior works) | Adaptive retrieval **only at semantic boundaries**           |
> | **Target Scenario**      | Primarily Long-Input                                         | **Diverse Long-Input, Long-Output Mixed Scenarios**          |
> | **Core Advantage**       | Achieves accurate KV retrieval for long inputs               | **Efficiently and accurately handles diverse long sequences (esp. long-output reasoning)** |

---

> ### Author Response · Authors · 2025-11-24
>
> **Q1. Additional Experimental Comparison with ClusterKV.**
>
> To substantiate the practical benefits of our innovations, we conduct new experiments to directly compare the inference efficiency and accuracy of LouisKV against ClusterKV on multiple tasks.
>
> - **Experimental Setup for Baselines:**
>
>   For a fair comparison, we adapt the official open-source implementation of ClusterKV to our experimental framework. We configured the ClusterKV baseline to adhere to its core principles: for both input and output sequences, KVs are clustered with an average cluster size of 16 and offloaded to CPU memory. For the output sequence, we set a GPU buffer of 512 tokens; clustering and offloading will be triggered when this buffer is full. ClusterKV’s retrieval operation is performed at every decoding step. The setup for LouisKV follows the description in our paper.
>
> - **End-to-End Inference Latency.**
>
>   We test on three representative long-sequence scenarios. The results clearly show that LouisKV significantly outperforms ClusterKV in short-input-long-output (500+16K) and long-input-long-output (32K+16K) scenarios, where ClusterKV’s per-token retrieval becomes a major bottleneck. As shown in the table below, the latency results indicate that LouisKV achieves up to an **18.8%** efficiency improvement over ClusterKV in long-output tasks. This clearly demonstrates that our adaptive retrieval strategy offers a significant advantage over ClusterKV's per-token retrieval approach when managing long output sequences.
>
>   | E2E Latency (s) | 32K+512 | 500+16K | 32K+16K |
>   | --------------- | ------- | ------- | ------- |
>   | ClusterKV       | 35.04   | 742.28  | 898.18  |
>   | LouisKV         | 31.9    | 634.79  | 729     |
>
> - **Inference Accuracy.**
>
>   We also compare accuracy on various long-input and long-output tasks. Specifically, we select datasets from LongBench for long-input understanding and sample 100 questions from each long-output reasoning task.
>
>   On long-input understanding tasks, both methods perform comparably. This is expected, as both employ a similar clustering strategy for the input sequence. However, on long-output reasoning tasks (e.g., MATH500), LouisKV demonstrates **significant accuracy advantages.** For example, it boosts accuracy on challenging datasets like MATH500 (0.86 vs 0.81) and AIME24 (0.66 vs 0.60). This is primarily attributed to our novel **dynamic segmentation** mechanism for the output sequence.
>
>   | Long-Input Tasks | Qasper | HotpotQA | GovReport | Avg. |
>   | ---------------- | ------ | -------- | --------- | ---- |
>   | ClusterKV        | 43.86  | 54.36    | 34.38     | 44.2 |
>   | LouisKV          | 44.31  | 55.64    | 33.9      | 44.6 |
>
>   | Long-Output Tasks | MATH500 | GPQA | AIME24 | LongReason-16K | LongReason-32K | LongReason-64K | Avg.  |
>   | ----------------- | ------- | ---- | ------ | -------------- | -------------- | -------------- | ----- |
>   | ClusterKV         | 0.81    | 0.62 | 0.6    | 0.62           | 0.63           | 0.61           | 0.648 |
>   | LouisKV           | 0.86    | 0.64 | 0.66   | 0.63           | 0.66           | 0.67           | 0.687 |
>
> **By combining these empirical results with our detailed discussion on novelty, we re-emphasize that LouisKV is not a similar work to ClusterKV, but a more efficient framework designed for broader and more complex scenarios.**

---

> ### Author Response · Authors · 2025-11-24
>
> **W2. Clarifying the Novelty of Adaptive Retrieval.**
>
> We agree with your point that the concepts of "exploiting temporal locality" and "using cosine similarity between query vectors" have been explored in prior works. **However, the novelty of our work lies in the UNIQUE** **application and purpose of these concepts.** The methods adopted in previous works leveraging this property and the outcomes they achieved differ significantly from ours.
>
> Previous works typically leverage the query similarity to **preload the KV cache of next layer** within a single decoding step to enhance inference efficiency. **In contrast, LouisKV innovatively applies this property to dynamically define the boundaries of KV segments in the output sequence.** This partitioning is the foundation of our output KV management and adaptive retrieval strategy, allowing us to trigger retrieval **per-segment rather than per-token**. This is a fundamentally new application of this concept that directly addresses the efficiency challenges of long-output scenarios, achieving an accuracy-efficiency win-win not realized by prior work.
>
> We believe these prior works are complementary to our approach rather than diminishing its novelty.
>
> **Conclusion**
>
> Thank you again for your valuable feedback. We hope our clarifications and the new experimental results have sufficiently demonstrated the novelty and contribution of LouisKV. **Our core contribution lies in proposing the first decoupled KV retrieval architecture, based on new observations of KV access patterns, that provides a unified and efficient solution for both long-input and long-output sequences.** We believe this work offers significant value to the community and kindly ask you to reconsider our submission.

---

### Official Review · Reviewer_14Z4 · 2025-11-04

**Soundness:** 3
**Presentation:** 3
**Contribution:** 2
**Rating:** 6
**Confidence:** 2

**Summary:**

The authors analyze long-sequence inference and discover that critical KVs (i.e. important tokens) demonstrate strong temporal locality during decoding (i.e., critical KVs between neighboring decoding steps are similar), and that critical KVs are sparsely distributed in prefill, but during decoding are locally concentrated within certain reasoning steps. Based on these observations, separate prefill/decoding cache management strategies are leveraged to improve recall from the CPU to the GPU of relevant KV entries. Custom kernels are implemented resulting in 4.7x speedup against baselines with minimal performance drop.

**Strengths:**

- Long-sequence KV strategies are very focused on long prefill. With the advent of reasoning models, the long decoding phase is now common, and the authors' study of this scenario is timely.

- The described approach is simple and easy to understand. Mixed CPU/GPU KV management strategies are relatively less common, but are becoming increasingly important as token eviction continues to prove less generalizable.

 - The method is both fast and maintains high performance on a wide variety of task types.

 - A variety of model families are tested thus demonstrating architecture generalizability.

**Weaknesses:**

- As with all mixed CPU/GPU KV cache strategies, the approach requires careful kernel level optimizations to mitigate data transfer latency, thus making this approach non-hardware/software agnostic.

- Selection of $\tau$ seems to vary by both model and task. Time spent tuning this hyper-parameter limits the overall efficiency of this framework.

- Small sample sizes from the math reasoning benchmarks.

**Questions:**

- Could you try more questions from MATH500 and GPQA? You do not need to try the entirety of these datasets, but perhaps try 2-3 50 question samples and report the error.

 - How quickly can $\tau$ can be tuned? That is, could you study how many questions are practically needed per model to determine a reasonable selection of this hyperparameter?

---

> ### Author Response · Authors · 2025-11-22
>
> Thanks for your review and valuable feedback. Your recognition of our strengths is a great encouragement, and the weaknesses and questions you raised are crucial for us to further improve our work.  We have conducted in-depth analysis and supplementary experiments based on your comments and would like to respond to each point below.
>
> **W1: Hardware/Software Agnostic.**
>
> We fully agree with your opinion that any mixed CPU/GPU KV cache strategy, to achieve maximum efficiency, inevitably requires low-level optimizations. We see this as a fundamental trade-off between performance and complete hardware agnosticism, which is common in high-performance systems research.
>
> Our core objective is to achieve **broad portability** within the most dominant high-performance computing ecosystem. Therefore, we directly utilize CUDA and Triton to take fruits from their well-tuned kernel-level optimizations. This ensures that LouisKV can be easily deployed across different NVIDIA GPUs and seamlessly integrated with mainstream deep learning frameworks like PyTorch. **Thus, while LouisKV is not fully hardware/software-agnostic, it possesses strong portability and practical utility in real-world applications.**
>
> **W2 & Q2: $ \tau $ Parameter Tuning Efficiency Study.**
>
> First, we would like to clarify a key point: **the $ \tau $ value is calibrated once per model, rather than per task.** Once an optimal $ \tau $ is determined for a model (e.g., Qwen3-8B), it is applied across all long-context tasks without repeated tuning.
>
> We empirically determine the optimal $ \tau $ via an efficient, one-time calibration process, identical to the methodology presented in Figure 8(c) of the paper. This involves evaluating a series of $ \tau $ values **on a small (~300 examples), diverse validation set** to map out the accuracy-latency trade-off curve. The $ \tau $ value corresponding to the curve's "knee point" is then selected, as it represents the best trade-off, offering high accuracy before latency starts to increase sharply.
>
> **W3 & Q1: Validation on larger MATH and GPQA datasets.**
>
> To more reliably validate our method's inference accuracy, we conduct larger-scale supplementary experiments as you suggest.  We sample 200 questions each from MATH and GPQA datasets, following the experimental setup from Section 5.2. The results are shown below:
>
> | Dataset | FullCache | Arkvale | LouisKV |
> | --- | --- | --- | --- |
> | **MATH500** | 88.5% | 85.0% | 87.5% |
> | **GPQA** | 62.5% | 58.0% | 61.0% |
>
> **The results demonstrate that LouisKV maintains accuracy close to FullCache on large-scale samples while outperforming Arkvale, the state-of-the-art KV cache retrieval method.**

---

> > ### Comment · Reviewer_14Z4 · 2025-11-24
> > **Thanks for the response**
> >
> > I thank the authors for their response, especially running additional experiments and clarifying $\tau.$ Since this is a primarily empirical work it would ultimately have been better to compare against a wider selection of eviction-based methods (SnapKV, PyramidKV, etc.)  and other SoTA offloading methods (MagicPIG, ShadowKV, InfiniGen). However, the results do indicate that there are clear accuracy and latency benefits to using LouisKV, so I will maintain my score and advocate for its acceptance.

---

### Author Response · Authors · 2025-12-03
**Summary of Rebuttal and Discussion for the Area Chair**

3. **Key Updates & Resolutions during Rebuttal.**

    We thank all reviewers for their constructive feedback. Below is a summary of how we addressed their specific concerns:

    1. **Resolved Core Concern on Novelty & Factual Misunderstanding (Reviewer dBjq)**

        Reviewer dBjq (Score 2) raised concerns that LouisKV overlaps with prior works like ClusterKV. We respectfully clarified that this concern arises from a factual misunderstanding.

        - **Misunderstanding:** The reviewer incorrectly stated that we use "fixed-size" segments for decoding.
        - **Clarification:** We clarified that LouisKV utilizes **Dynamic Variable-Length Segmentation** driven by semantic boundaries, which is fundamentally different from ClusterKV's approach. We also highlighted our **Decoupled Management Scheme** (Clustering for Input vs. Dynamic Segmentation for Output).
        - **Experimental Evidence:** We added a direct comparison with ClusterKV. Results demonstrate that LouisKV achieves higher accuracy on reasoning tasks (e.g., MATH500: 0.86 vs. 0.81) and lower latency (up to 18.8% faster) due to our adaptive strategy, proving our distinct contribution.
    2. **Validated Generalizability & Tuning Efficiency (Reviewer 14Z4)**
        - **Expanded Validation:** In response to concerns about sample size, we expanded our evaluation on MATH500 and GPQA to 200 samples. The results confirmed that LouisKV 的 accuracy consistently outperforming SOTA methods like Arkvale.
        - **Tuning Clarification:** We clarified that the hyperparameter $\tau$ requires only **one-time calibration per model** (not per task), ensuring practical deployment efficiency.
    3. **Demonstrated Robustness on Long-Context & System Analysis (Reviewer cVfv)**
        - **RULER Stress Test:** We conducted stress tests on the RULER benchmark (32K-128K). LouisKV successfully handled 128K contexts (where FullCache failed due to OOM) and consistently outperformed Arkvale.
        - **System Analysis:** We provided comprehensive analyses on **clustering overhead** (showing negligible impact on TTFT due to asynchronous pipelining and Triton-optimized kernel), **trigger statistics** (validating the necessity of layer-wise segmentation), and **data traffic** (demonstrating a **77% reduction** in CPU-GPU traffic compared to per-token methods).
        - **Cluster Locality:** We analyzed the spatial distribution of clusters and confirmed that they aggregate widely dispersed tokens throughout the entire context instead of forming contiguous spans.
    4. **Clarified Conceptual Novelty & Parameter Sensitivity (Reviewer 7kuj)**
        - **Conceptual Novelty:** We elucidated that LouisKV’s core contribution is the first unified and efficient KV retrieval framework derived from novel observations of distinct KV distribution patterns tailored for diverse long-sequence scenarios, rather than a mere combination of existing techniques.
        - **Sensitivity Analysis:** We provided the sensitivity analysis for Sink Tokens ($S$), Local Buffer size ($W$), and Cluster Counts. The results demonstrate that our method remains stable and robust across reasonable hyperparameter ranges.
        - **Overhead Quantification:** We quantified the clustering latency, confirming it is effectively hidden by the prefill computation.

We believe the rebuttal process has effectively resolved the core concerns raised by the reviewers, resulting in a significantly strengthened manuscript. **We strongly recommend reviewing the detailed rebuttal responses**, as they detail our clarifications on novelty and the extensive empirical validation. We are confident that the revised manuscript meets the standards of ICLR and respectfully request the Area Chair to consider these substantial improvements in the final assessment.

Best regards,

The Authors of Submission #11436

---

### Author Response · Authors · 2025-12-03
**Summary of Rebuttal and Discussion for the Area Chair**

Dear Area Chair:

We sincerely thank the reviewers (14Z4, dBjq, cVfv, 7kuj) for their exhaustive feedback. These insights prompted us to conduct extensive additional experiments and deep analyses, which have significantly enhanced the completeness and contribution of our work.

1. **Engagement Status during Rebuttal.**

    We are very grateful to **reviewer 14Z4**, who responded promptly, acknowledged our new experiments, and confirmed their support for the acceptance of our paper.  Unfortunately, reviewers dBjq, cVfv, and 7kuj were unable to respond before they could provide any further feedback, as the discussion phase have already ended. **However, we believe we have resolved the core concerns of all reviewers and kindly request the AC to consider our detailed responses when making the final decision.**

2. **Summary of Consensus.**

    Despite varying final scores, there was a consensus on the following strengths:

    - **Sound Observation and Motivation:** Reviewers acknowledged our key observations regarding strong temporal locality and distinct input/output KV distributions, agreeing that they provide a sound and insightful foundation for the proposed framework.
    - **Effective Design and Solid System work:**  Reviewers acknowledged the efficiency of our semantic-aware adaptive KV retrieval strategy and decoupled KV management scheme. Furthermore, they recognized the value of our kernel-level optimizations.
    - **Comprehensive Evaluation and Strong Result:** Reviewers recognized the extensive experiments covering diverse long-sequence scenarios, including challenging long-output reasoning tasks (e.g., MATH500 and GPQA). They confirmed that LouisKV achieves superior speedups with near-lossless accuracy compared to SOTA methods.

---

### Meta-Review · Area_Chair_6jTJ · 2025-12-23

**Summary:**

Reviewers unanimously acknowledged the validity of the research observations and motivations, the efficiency of the methodological design, and the comprehensiveness of the experiments. Core concerns centered on: methodological innovation (distinction from existing work such as ClusterKV), generalization capability and tuning efficiency, robustness to long context and system overhead, as well as parameter sensitivity and the explanation of conceptual novelty. The authors actively addressed these core issues through supplementary experiments, clarification of misunderstandings, and quantitative analysis.

**Reviewer Concerns:**

Reviewer 14Z4's initial rating was 6. They acknowledged the authors' supplementary experiments and clarifications, further solidifying their supportive stance and maintaining a high score.

Reviewer dBjq's initial rating was 2, but did not respond to the authors' rebuttal. I believe the authors' rebuttal clarified the misunderstanding and added comparative experiments to validate innovation, warranting an increased score.

Reviewer cVfv's initial rating was 6, but did not respond to the authors' rebuttal. The authors addressed core concerns regarding long-context robustness and supplementary systematic analysis; the rating should remain positive for acceptance.

Reviewer 7kuj's initial rating was 4, but did not respond to the authors' rebuttal. I believe the issues concerning conceptual novelty and parameter sensitivity were sufficiently clarified; the rating should be upgraded.

**Reviewer Scores:**

I think they will maintain or increase the rating to positive acceptance.

---

### Decision · Program_Chairs · 2026-01-26

Accept (Poster)